# EMO: Earth Mover Distance Optimization for Auto-regressive Language Modeling

**Siyu Ren**[12][*]**, Zhiyong Wu**[2][†]**, Kenny Q. Zhu**[3][†]
[1]Shanghai Jiao Tong University [2]Shanghai AI Laboratory [3]University of Texas at Arlington
`roy0702@sjtu.edu.cn, wuzhiyong@pjlab.org.cn, kenny.zhu@uta.edu`

## Abstract

Neural language models are predominantly trained using maximum likelihood estimation (MLE), which is equivalent to minimizing the forward cross-entropy between the empirical data distribution and the model distribution. However, various degeneration phenomena are still widely observed when decoding from the distributions learned by such models. We establish that the forward cross-entropy is suboptimal as a distance metric for aligning human and model distribution due to its (1) recall-prioritization (2) negative diversity ignorance and (3) train-test mismatch. In this paper, we propose **E**arth **M**over Distance **O**ptimization (EMO) for auto-regressive language modeling. EMO capitalizes on the inherent properties of earth mover distance to address the aforementioned challenges. Due to the high complexity of direct computation, we further introduce a feasible upper bound for EMO to ease end-to-end training. Upon extensive evaluation, EMO demonstrates a consistently better language modeling performance than MLE across domains. Moreover, EMO shows noteworthy enhancements in downstream performance with minimal fine-tuning on merely 25,000 sentences, highlighting its potential as a lightweight calibration method for enhancing large-scale pre-trained language models. Code available at `https://github.com/DRSY/EMO`.

## 1 Introduction

The dominant paradigm of natural language generation systems hinges on probabilistic neural language models (Radford et al., 2019; Zhang et al., 2022), which permit evaluating the probability of any given text sequence as well as generating novel ones using various decoding strategies upon learned distributions (Holtzman et al., 2019; Meister et al., 2023b). Language modeling, the process of aligning model distribution with that of human language, is usually formulated as a sequence prediction task in which maximum likelihood estimation (MLE) is typically adopted as the training objective owing to its simplicity and intuitiveness.

However, various text degeneration phenomena with incoherent and nonsensical (LeBrun et al., 2021; Holtzman et al., 2019) content are still widely observed in text generated from language models pre-trained on massive amounts of human data. This indicates that the model distribution $Q_\theta$ (parametrized by $\theta$) learned by MLE still differs substantially from the human language distribution $P$, despite having a seemingly low training loss (Meister et al., 2023b). From a distributional view, training with MLE is equivalent to minimizing the *forward cross-entropy* between $P$ and $Q_\theta$:

$$\text{CE}(P, Q_\theta) = -\mathbb{E}_{x \sim P}[\log Q_\theta(x)] \tag{1}$$

We argue that the forward cross-entropy has inherent limitations as a metric for matching model distribution and that of human language. Firstly, forward cross-entropy is recall-prioritized (Meister et al., 2023a). At each time step, it focuses exclusively on increasing the model likelihood of the ground-truth next token. This can result in poor precision of the learned model distribution when training data is noisy or slow convergence even when sufficient amounts of high-quality text corpus are available. Secondly, when used in language model pre-training, forward cross-entropy faces

---

[*]Work done during an internship at Shanghai AI Laboratory.

[†]Correspondence to: Zhiyong Wu, Kenny Q. Zhu, partly supported by National Science Foundation (NSF) Grant No. 2349713, and University of Texas STARs program.

the negative diversity ignorance issue (Li et al., 2019) where all non-ground-truth next tokens are deemed as equally incorrect. However, some tokens might be less incorrect or even plausible alternatives to the ground truth than other tokens. Capturing these latent negative diversity can assist language models in enhancing their modeling of the human language distribution. Thirdly, the form of forward cross-entropy is inconsistent with how language models are evaluated (Pang & He, 2020). Such a train-test objective mismatch makes MLE a less reliable indicator of modeling quality.

To alleviate the aforementioned limitations of MLE, we direct our attention towards an alternative distance metric, namely the Earth Mover Distance (EMD) (Kantorovich, 1960). EMD is initially discussed in the context of optimal transport problem (Villani et al., 2009) and then incorporated as a distance metric for implicit generative modeling, e.g., WGAN (Arjovsky et al., 2017) and WAE (Tolstikhin et al., 2018). The appeal of EMD lies in (1) it takes into account both precision and recall during modeling; (2) it acknowledges the varying degrees of correctness in data samples, enabling more nuanced training signals. (3) its mathematical formulation permits better consistency between the training and testing phases. Given these properties, we incorporate EMD as a better token-level probability distance measure into language modeling. However, computing the exact value of EMD requires external solvers that are detached from the computation graph and block gradient back-propagation. We overcome this issue by developing an differentiable upper bound of EMD (DEMD) that can be optimized in an end-to-end manner without resorting to external specialized solvers. Combined with a semantically informed transport cost function, we present EMO (**E**arth **M**over Distance **O**ptimization) for training auto-regressive language models.

We first evaluate the effectiveness of the proposed method on the task of language modeling across diverse domains and show that EMO yields generations with significantly higher distributional closeness (6.2 points on average measured by MAUVE) with human text. We further demonstrate that, by applying EMO in a lightweight fine-tuning stage using several orders of magnitude fewer tokens than pre-training, pre-trained LLMs' performance on a range of downstream language understanding tasks can be significantly boosted, e.g., an average improvement of 4 points across 8 datasets. By progressively increasing the volume of data utilized for continual fine-tuning, EMO also demonstrates superior scaling properties compared to existing methods.

## 2 BACKGROUND AND MOTIVATION

### 2.1 AUTO-REGRESSIVE LANGUAGE MODELING

Current language generation systems are predominantly based on probabilistic neural auto-regressive language models (LMs) (Bengio et al., 2000). Denoting a language model parametrized by $\theta$ as $Q_\theta$, it essentially computes the probability of a given text sequence $\boldsymbol{x}$ as the product of each token's conditional probability given preceding tokens:

$$Q_\theta(\boldsymbol{x}) = \prod_{t=1}^{|\boldsymbol{x}|} Q_\theta(x_t|x_{<t}) \tag{2}$$

Prior works have adopted various neural architectures, e.g., LSTM (Hochreiter & Schmidhuber, 1997), GRU (Chung et al., 2014), and now the most widely used Transformer (Vaswani et al., 2017), to transform natural language input to next token probability. To estimate $\theta$, the most common approach is to perform self-supervised pre-training on enormous volume of text corpus using maximum likelihood estimation. Given the data distribution $P$, the training objective of MLE is equivalent to minimizing the forward cross-entropy between $P$ and $Q_\theta$:

$$\mathcal{L}_{\text{MLE}} = \text{CE}(P, Q_\theta) = -\mathbb{E}_{\boldsymbol{x} \sim P}[\log Q_\theta(\boldsymbol{x})] \tag{3}$$

$$= -\mathbb{E}_{\boldsymbol{x} \sim P}[\sum_{t=1}^{|\boldsymbol{x}|} \log Q_\theta(x_t|x_{<t})] \tag{4}$$

$$= \mathbb{E}_{x \sim P}[\sum_{t=1}^{|\boldsymbol{x}|} \text{CE}(P(\cdot|x_{<t}), Q_\theta(\cdot|x_{<t}))] \tag{5}$$

From Eq. 3 to Eq. 5, the sentence-level cross-entropy is further decomposed into the sum of forward cross-entropy between token-level data distribution $P(\cdot|x_{<t})$ and model distribution $Q_\theta(\cdot|x_{<t})$.

## 2.2 Deficiency of Maximum Likelihood Estimation

In this subsection, we will delve into certain properties of the forward cross-entropy employed in MLE training, and elucidate the impact of these properties on the learned model distribution.

### 2.2.1 Recall-Prioritization

A series of recent works (Lucic et al., 2018; Sajjadi et al., 2018; Djolonga et al., 2020) have generalized the classification metric *recall* to measure the quality of generative modeling. Specifically, recall here is defined as the model distribution $Q_\theta$'s coverage of data distribution $P$, i.e., a high recall means that high likelihood tokens under $P$ shall also have high likelihood under $Q_\theta$. In contrast, the *precision* of $Q_\theta$ focuses on measuring whether low-quality tokens (unlikely under $P$) have low probabilities under $Q_\theta$. To elaborate further, we derive the gradient of forward cross-entropy w.r.t model parameters $\theta$ as follows:

$$\nabla_\theta \mathcal{L}_{\text{MLE}} = -\mathbb{E}_{\boldsymbol{x} \sim P}[\sum_{t=1}^{|\boldsymbol{x}|} \frac{\nabla_\theta Q_\theta(x_t|x_{<t})}{Q_\theta(x_t|x_{<t})}] \tag{6}$$

Eq. 6 clearly shows that, by minimizing $\mathcal{L}_{\text{MLE}}$ via gradient descent, $Q_\theta$ is encouraged to only assign a high probability to the ground-truth next token and therefore being recall-prioritized. Consequently, the precision of $Q_\theta$ is not adequately incentivized in MLE because Eq. 6 does not explicitly discourage learning of low-quality tokens. In short, recall-prioritization results in insufficient optimization of $Q_\theta$'s precision and amplifies the need for enormous amounts of high-quality text corpus to overcome this limitation.

### 2.2.2 Negative Diversity Ignorance

Another noteworthy property of MLE is its ignorance of diverse supervision signals of non-ground-truth tokens during auto-regressive language modeling Zhang & Zhao (2018); Li et al. (2019). Specifically, MLE assumes token $x_t$ observed in training sample $\boldsymbol{x}$ is the only ground-truth token at time step $t$ and maximizes its log-likelihood under $Q_\theta$. Concurrently, the remaining tokens other than $x_t$ in the vocabulary are treated *equally incorrect*, and their probabilities are implicitly penalized in MLE. This can be demonstrated by analyzing the partial derivative of $\text{CE}(P(\cdot|x_{<t}), Q_\theta(\cdot|x_{<t}))$ w.r.t the output logits $\boldsymbol{z}$ before softmax:

$$\frac{\partial \text{CE}(P(\cdot|x_{<t}), Q_\theta(\cdot|x_{<t}))}{\partial z_i} = \begin{cases} Q_\theta(x_t) - 1 & \text{if } v_i = x_t \\ Q_\theta(v_i) & \text{others} \end{cases} \tag{7}$$

where $v_i$ denotes the $i$-th token in the vocabulary. To reach a local minimum during gradient-based optimization (gradient norm $\to 0$), the model will try to increase the probability of $x_t$ ($Q_\theta(x_t) \to 1$) and decrease the probability of all other tokens ($Q_\theta(v_i) \to 0$). In practice, however, certain tokens can serve as plausible alternatives to $x_t$, e.g., synonyms of $x_t$. The training objective should assign high probabilities to those tokens rather than penalize them as did in MLE. In essence, such an inability of MLE may inhibit building more powerful neural models of human language that can accurately distinguish the relative correctness of the next token.

### 2.2.3 Train-test mismatch

When training is completed, language models are often evaluated against objectives that differ significantly from MLE. For example, ROUGE (Lin, 2004) for summarization and BLEU (Papineni et al., 2002) for machine translation. This creates a train-test mismatch for language modeling. In other words, we draw sample $\boldsymbol{x}$ from $Q_\theta$ and then assess its quality using certain evaluation function $f(\cdot)$, i.e., maximizes $\mathbb{E}_{\boldsymbol{x} \sim Q_\theta}[f(\boldsymbol{x})]$, where $f(\cdot)$ varies according to different downstream scenarios. This is inconsistent with MLE in which the expectation is taken w.r.t data distribution $P$, i.e., $\mathbb{E}_{x \sim P}[\log Q_\theta(x)]$. Most prior works have attempted to address this issue by incorporating the evaluation objective $f(\cdot)$ into training and adopting reward-augmented maximum likelihood Norouzi et al. (2016); Zhang & Zhao (2018); Liu et al. (2022) based on the policy gradient theorem (Sutton et al., 1999) or contrastive learning. However, such changes incur non-trivial overhead, and the choices of evaluation function $f(\cdot)$ are usually task-specific and less applicable for general language modeling. In light of this, there is a critical need for objectives that exhibit better train-test consistency to enhance the efficacy of language modeling.

## 3 EMO: EARTH MOVER DISTANCE OPTIMIZATION

In pursuit of a divergence measure that circumvents the adverse properties of forward cross-entropy, we draw our attention to the Earth Mover's Distance (EMD), a distance function that was originally studied in the context of optimal transport planning of goods and materials (Kantorovich, 1960; Villani, 2021) and then borrowed for generative modeling by ML community (Arjovsky et al., 2017; Tolstikhin et al., 2018). In Section 3.1, we provide the formal definition of EMD and elucidate its adaptation for auto-regressive language modeling with a semantically informed cost function. In Section 3.3, we tackle the challenge posed by the intractable infimum associated with EMD by developing its upper bound. Collectively, we introduce EMO, an approach dedicated to the training of auto-regressive language models through the optimization of the Earth Mover's Distance.

### 3.1 ADAPTING EARTH MOVER'S DISTANCE TO AUTO-REGRESSIVE LANGUAGE MODELING

Formally, given two probability distributions $P_1$ and $P_2$ over a metric space $\mathcal{X}$, the earth mover's distance between $P_1$ and $P_2$ is defined as the minimum accumulative cost of moving all probability mass of $P_1$ to $P_2$:

$$\text{EMD}(P_1, P_2) = \inf_{\gamma \in \Pi(P_1, P_2)} \mathbb{E}_{(x_1, x_2) \sim \gamma}[C(x_1, x_2)] \tag{8}$$

where inf stands for infinitesimal, $\Pi(P_1, P_2)$ denotes the set of all joint distributions $\gamma(x_1, x_2)$ whose marginals are $P_1$ and $P_2$, respectly. $\gamma(x_1, x_2)$ is interpreted as the amount of probability mass transported from $P_1(x_1)$ to $P_2(x_2)$. $C(x_1, x_2)$ is a non-negative function that measures the cost of transporting a unit mass from $x_1$ to $x_2$. In the context of auto-regressive language modeling, $P_1$ refers to the model distribution to be learned and $P_2$ refers to the data distribution, both representing the locally factorized probability distribution over the next token at time step $t$ given preceding tokens, i.e., $P_1 := Q_\theta(\cdot|x_{<t})$ and $P_2 := P(\cdot|x_{<t})$. Thus, Eq. 8 can be reformulated as:

$$\text{EMD}(Q_\theta(\cdot|x_{<t}), P(\cdot|x_{<t})) = \inf_{\gamma \in \Pi(Q_\theta(\cdot|x_{<t}), P(\cdot|x_{<t}))} \mathbb{E}_{(x_1, x_2) \sim \gamma}[C(x_1, x_2)]$$

$$= \inf_{\gamma \in \Pi(Q_\theta(\cdot|x_{<t}), P(\cdot|x_{<t}))} \sum_{i=1}^{|V|} \sum_{j=1}^{|V|} \gamma(v_i, v_j) C(v_i, v_j) \tag{9}$$

where $V$ is the vocabulary of language model and $v_i$ indexes the $i$-th token in $V$. Once the cost function $C$ is defined, computing the above earth mover's distance amounts to solve the following constrained linear optimization problem:

$$\min_\gamma \sum_{i=1}^{|V|} \sum_{j=1}^{|V|} \gamma(v_i, v_j) C(v_i, v_j) \tag{10}$$

$$s.t \quad \sum_{j=1}^{|V|} \gamma(v_i, v_j) = P(v_i|x_{<t}) \quad \forall i \in \{1, ..., |V|\}$$

$$\sum_{i=1}^{|V|} \gamma(v_i, v_j) = Q_\theta(v_j|x_{<t}) \quad \forall j \in \{1, ..., |V|\}$$

**Semantically-Informed Transport Cost** The next step is to establish a definition of $C$ such that it reflects a meaningful distance between pairs of tokens $v_i$ and $v_j$. Intuitively, tokens that are more likely to be used interchangeably should have smaller distances, e.g., *glad* and *happy*. Conversely, tokens that are improbable to fit within each other's context, e.g., *cat* and *galaxy*, should be farther away. One such measure of token distance is naturally provided by their cosine distance in the contextual embedding space, i.e., $C(v_i, v_j) = 1 - \frac{e_i^\top e_j}{|e_i||e_j|}$, where $e_i$ is the $i$-th column of the language modeling head $\boldsymbol{E}$ of a LM $Q_\phi$ pre-trained via MLE. Because during training $e_i$ is optimized to be close to the contextual representation of all prefixes of which the next token is $v_i$, the cosine distance between $e_i$ and $e_j$ therefore serves as an effective proxy for quantifying the transport cost between $v_i$ and $v_j$. Once initialized, $\boldsymbol{E}$ will be jointly updated with other model parameters $\theta$ so that $C$ can capture the shift in the domain-specific distribution from the pre-training corpus.

## 3.2 A TRACTABLE UPPER BOUND

The complexity of traditional EMD solvers (Ling & Okada, 2007; Shirdhonkar & Jacobs, 2008) for computing Eq.10 is $O(|V|^3 \log |V|)$, which becomes burdensome for recent LLMs whose vocabulary can contain several tens of thousands of tokens. Additionally, employing external solvers disrupts gradient propagation, making end-to-end training infeasible. To tackle these challenges, we present a tractable upper bound of EMD that allows for efficient gradient-based optimization.

We start by defining a transport plan $\tilde{\gamma}$ that directly leverages the data distribution $P(\cdot|x_{<t})$ and model distribution $Q_\theta(\cdot|x_{<t})$ meanwhile being valid by adhering to the constraints stated in Sec. 3.1:

$$\tilde{\gamma}(v_i, v_j) = Q_\theta(v_i)P(v_j) \tag{11}$$

Here we omit the prefix $x_{<t}$ for notational simplicity. Essentially, $\tilde{\gamma}$ represents the probability of a data-dependent transport plan that moves the probability mass of $v_i$ under $Q_\theta$ to other tokens according to the proportions specified by $P$. Since both $Q_\theta$ and $P$ add up to 1, $\tilde{\gamma}$ is therefore a legitimate but not necessarily optimal plan. Denoting the unknown optimal plan with minimal transport cost as $\gamma^*$, we have the following inequality holds:

$$\text{EMD}(Q_\theta, P) \leq \sum_{i=1}^{|V|} \sum_{j=1}^{|V|} \tilde{\gamma}(v_i, v_j) C(v_i, v_j)$$

$$= \sum_{i=1}^{|V|} \sum_{j=1}^{|V|} Q_\theta(v_i) P(v_j) C(v_i, v_j) \tag{12}$$

$$= Q_\theta^\top C P \tag{13}$$

$$= Q_\theta^\top (\mathbf{1}\mathbf{1}^\top - \hat{\boldsymbol{E}}^\top \hat{\boldsymbol{E}}) P$$

$$= 1 - (\hat{\boldsymbol{E}} Q_\theta)^\top \hat{\boldsymbol{E}} P \tag{14}$$

where $\boldsymbol{C} \in \mathbb{R}^{|V| \times |V|}$ is the matrix notation of $C(v_i, v_j)$ used to transform the summation (Eq. 12) into quadratic form (Eq. 13), $\mathbf{1}$ is a all-one column vector, $\hat{\boldsymbol{E}}$ is the row-wise normalized version of $\boldsymbol{E}$, and $P$ is the one-hot next token distribution. Prior works on distribution matching using EMD Arjovsky et al. (2017); Gulrajani et al. (2017) commonly resort to the Kantorovich-Rubinstein duality (Villani, 2008) or entropic regularization (Cuturi, 2013; Frogner et al., 2015), which either conduct adversarial training of the generative model with an additional 1-Lipschitz critic network or adopt Sinkhorn-like iteration algorithm. In contrast, the upper bound we derived above only pertains to the training of $Q_\theta$, therefore being more stable and efficient for optimization. We term Eq. 14 as DEMD and incorporate it in conjunction with MLE (A.3) for auto-regressive language modeling.

**Generalized Form for Arbitrary** $P$ When $P$ is dense, the optimal solution of Eq. 14 is a one-hot distribution with all probability mass placed on the token with the smallest expected transport cost, rather than $P$. To tackle this, we derive the following generalized form for arbitrary $P$, which minimizes the absolute difference between the surrogate transport cost of $Q_\theta$ and $P$:

$$\widetilde{\text{DEMD}}(Q_\theta, P) = |Q_\theta^\top - P^\top| \boldsymbol{C} P \geq |Q_\theta^\top \boldsymbol{C} P - P^\top \boldsymbol{C} P| \tag{15}$$

## 3.3 BEHAVIORAL DIFFERENCES COMPARED TO MLE

Next, we delve into some properties of the proposed DEMD and provide insights on how it improves over MLE in terms of behavioral differences during optimization. To begin with, we first present DEMD's gradient with respect to model parameters $\theta$ (assuming a one-hot $P$):

$$\nabla_\theta \text{DEMD}(Q_\theta, P) = \sum_{i=1}^{|V|} \nabla_\theta Q_\theta(v_i) (\sum_{j=1}^{|V|} P(v_j) C(v_i, v_j)) = \sum_{i=1}^{|V|} \nabla_\theta Q_\theta(v_i) \mathbb{E}_{v_j \sim P}[C(v_i, v_j)] \tag{16}$$

**Harmonizing Recall and Precision** MLE is shown to be recall-prioritizing in the sense that its gradient update only ensures the target token is assigned with high probability. As a result, MLE-induced model tends to be over-confident on the low-quality regions in human language. In contrast, at each time step, DEMD also takes into account the precision of $Q_\theta$ by explicitly penalizing

low-quality tokens, i.e., those tokens will have large transport costs and thus large penalties. By effectively alleviating the overestimation of degenerated text, EMO better operationalize the harmonization of recall and precision compared to MLE.

**Negative Diversity Awareness**    The awareness of diverse supervisory signals of all tokens naturally arises from the recall-precision balancing property of DEMD. From Eq. 16 we can see that, the update of model parameters $\theta$ in DEMD comprises the sum of the gradients of the model's token probabilities across the entire vocabulary, weighted by their expected transport cost. Specifically, by employing gradient descent, tokens that deviate significantly from the data distribution (resulting in higher transport costs) will be down-weighted more severely than tokens that are contextually similar to the data distribution. Thus, the model distribution $Q_\theta$ learns to allocate probability mass more accurately than MLE due to the availability of more informative training signals.

**Better Train-Test Consistency**    One notable downside of forward cross-entropy is its train-test disparity nature (Sec. 2.2.3). Namely, during the training phase, its objective involves an expectation that is computed with respect to the data distribution $P$, whereas during testing, samples are drawn from the model distribution $Q_\theta$ and evaluated by humans. By rewriting Eq.12 as $\mathbb{E}_{v_i \sim Q_\theta}[\sum_{j=1}^{|V|} P(v_j)C(v_i, v_j)]$, we can see that DEMD explicitly involves the optimization of the expected transport cost computed with respect to $Q_\theta$. Therefore, DEMD has a higher degree of train-test consistency compared to MLE.

## 4    EXPERIMENT

We demonstrate EMO's empirical performance as a continual fine-tuning method upon pre-trained LMs. In Sec. 4.1, we compare EMO against MLE as well as other training criteria on a diverse range of language modeling datasets. In Sec. 4.2, we investigate the effectiveness of EMO on natural language understanding tasks under the few-shot in-context learning setting based on LLMs with various scales. The evaluation of EMO in instruction-tuning scenario is deferred to Appendix C.

### 4.1    LANGUAGE MODELING

#### 4.1.1    SETUP

**Task Definition and Evaluation Metric**    To gauge the quality of the learned model distribution after fine-tuning on a domain-specific corpus, we provide the model with a prefix and request it to continue with a segment of text that should ideally be similar to the reference text. We adopt Mauve (Pillutla et al., 2021) as the main evaluation metric, which compares the generated continuation against human text by calculating the area under the KL divergence curve and has seen wide usage in open-ended text generation (Ji et al., 2022; Zhang et al., 2023; Meister et al., 2023a).

**Pre-trained Language Models**    We utilize two representative decoder-only Transformer (Vaswani et al., 2017) language models, namely GPT-2 (Radford et al., 2019) and OPT-125M (Zhang et al., 2022), as $Q_\theta$, and fine-tune them using distinct training criteria including EMO and several recently proposed methods discussed below.

**Baselines**    Aside from MLE, we also compare with the following baselines, which aim to address the shortcomings of MLE by introducing novel divergence measures (or approximated variants) as their training objectives: (1) TaiLr (Ji et al., 2022) adopts the total variation distance (Van Handel, 2014) as a more robust measure between probability distributions and uses its token-level factorization as the training objective. (2) MixCE (Zhang et al., 2023) penalizes low-quality samples by leveraging an approximated version of reverse cross-entropy. For TaiLr and MixCE, we follow the implementations from their corresponding official codebase. More discussions regarding these baselines can be found in the Appendix A.1.

**Datasets**    We use 6 English textual corpora from 5 different domains for comprehensive evaluation:(1) WikiText-2 and WikiText-103 (Merity et al., 2016) are two commonly used language modeling benchmarks consisting of high-quality Wikipedia articles. (2) WebText$_{\text{test}}$ (Radford et al.,

2018) is the test set of the official WebText dataset from OpenAI, that was used to train GPT-2. (3) Penn Tree Bank (PTB) (Marcus et al., 1993) contains Wall Street Journal material in financial domain. (4) WritingPrompts (Fan et al., 2018) features text from the writing prompts forum of Reddit. (5) AG News (Zhang et al., 2015) is a collection of news articles from diverse domains, e.g., business, sports, and science. The statistics of each dataset are deferred to the Appendix A.4.

**Training Details**   We fine-tune GPT-2 and OPT-125M for 3 epochs on the training set of each dataset and save the model checkpoint with the lowest validation loss. We use the AdamW (Loshchilov & Hutter, 2018) optimizer with a learning rate of 5e-5. The batch size is fixed as 32 for all experiments. The maximum input length during training is set to 256. For TaiLr and MixCE that involve weighting coefficient, we conduct a hyperparameter sweep within $\{0.9, 0.8, 0.7\}$. EMO does not necessitate any hyperparameter tuning.

**Decoding Algorithm**   To gauge the quality of the learned model distribution $Q_\theta$ in a faithful way (Eikema & Aziz, 2020), we employ unbiased sampling (also known as ancestral sampling) as the primary decoding algorithm throughout the experiments. The length of prefixing and generated tokens for each dataset can be found in Appendix A.4. We repeat the sampling process 5 times for each prefix and report the average Mauve score.

### 4.1.2   MAIN RESULTS

Table 1: Unbiased sampling results (Mauve↑) of models fine-tuned by EMO as well as compared baselines. Numbers are the mean of 5-run sampling, aggregated over 3 different random seeds. **Bold** numbers indicate the results are significantly better than MLE with $p$-value $< 0.001$.

| Model | Objective | WikiText2 | WikiText103 | WebText$_{\text{test}}$ | PTB | WritingPrompts | AG |
|-------|-----------|-----------|-------------|-------------------------|-----|----------------|-----|
| GPT-2 | MLE | 77.5 | 77.1 | 75.5 | 76.1 | 83.6 | 75.0 |
| | TaiLr | 79.6 | 78.0 | 76.5 | 73.8 | 84.1 | 75.8 |
| | MixCE | 78.3 | 77.6 | 76.3 | 76.9 | 82.7 | 76.6 |
| | EMO | **87.5** | **82.1** | **80.5** | **79.6** | **87.4** | **84.9** |
| OPT$_{125M}$ | MLE | 77.2 | 75.8 | 74.7 | 83.6 | 84.1 | 82.1 |
| | TaiLr | 78.4 | 75.2 | 74.2 | 82.2 | 83.4 | 81.8 |
| | MixCE | 78.6 | 75.4 | 75.3 | 81.5 | 83.5 | 83.2 |
| | EMO | **82.9** | **81.0** | **80.7** | **86.1** | **87.9** | **84.8** |

Table 1 summarizes the unbiased sampling results of GPT-2 and OPT-125M fine-tuned with different training objectives on six datasets. We can clearly observe that EMO consistently outperforms MLE and other recently proposed training criteria across various domains. Although TaiLr and MixCE both leverage new distance measures that have theoretical advantages over forward cross-entropy, they suffer from either a mild assumption about the model's training dynamics or degeneration into a regularized version of forward cross-entropy. Therefore, they still exhibit the same drawbacks of MLE stated in Sec. 2.2. In contrast, EMO effectively manifests its theoretical advantages and leads to language models with more human-like distribution. For more quantitative results about the learned model distribution please refer to Appendix A.4.1.

### 4.1.3   EXPERIMENT WITH ORACLE DATA GENERATOR

In addition to the setting where we only have access to training data sampled from unknown distribution, in this subsection we seek to analyze more fine-grained distributional properties of models trained with different criteria. Specifically, we use training data sampled from an orcale GPT-2-Large model whose distribution $P$ is known. We use the GPT-2-output dataset consisting of 250,000/5,000/5,000 paragraphs in the training/validation/test set generated via unbiased sampling.

**Setup**   Apart from Mauve for measuring the sequence-level similarity between texts sampled from the learned and oracle model, we also incorporate model's test set perplexity PPL$_{\text{test}}$ , the oracle model's perplexity PPL$_{\text{oracle}}$ (calculated using oracle model on model-generated texts) and ROUGE-1/L (Lin, 2004) that evaluate the learned distributions from different perspectives. PPL$_{\text{test}}$ is commonly adopted as a quantitative measure of model's *recall*. PPL$_{\text{oracle}}$ emphasizes more on *precision*

by penalizing generations $\boldsymbol{x}$ from $Q_\theta$ that are unlikely to be produced by the oracle model $P$, i.e., $P(\boldsymbol{x})$ is low, While ROUGE score focuses on *recall* by rewarding high n-gram overlap. For each training method, we fine-tune GPT-2 using the same experimental setting as described in Sec. 4.1.1.

Table 2: Unbiased sampling results of GPT-2 fine-tuned with different training criteria. Numbers are the mean of 5-run sampling, aggregated over 3 different random seeds. **Bold** numbers indicate the results are significantly better with $p$-value $< 0.001$.

| Methods | PPL$_{\text{test}}$ ↓ | PPL$_{\text{oracle}}$ ↓ | Mauve↑ | ROUGE-1↑ | ROUGE-L↑ |
|---------|------|------|------|------|------|
| MLE | **70.1** | 114.46 | 77.5 | 34.59 | 29.85 |
| TaiLr | 73.5 | 95.22 | 77.4 | 34.95 | 30.09 |
| MixCE | 74.4 | 79.46 | 78.4 | 35.31 | 30.26 |
| EMO | 74.9 | **55.85** | **83.4** | **37.37** | **31.17** |

**Results**  We report the performance of EMO as well as baseline methods in Table 2. The results consistently reveal that EMO outperforms all baseline methods across all evaluation metrics, except for PPL$_{\text{test}}$. Notably, EMO exhibits a significantly reduced PPL$_{\text{oracle}}$ compared to the baseline methods, demonstrating its effective mitigation of the overestimation issue associated with low-quality text in prior divergence measures. The awareness of diversity within the range of plausible tokens in addition to the gold token is naturally reflected in EMO's higher PPL$_{\text{test}}$. As indicated by the highest MAUVE score, EMO strikes the best balance between recall and precision, suggesting that utilizing a well-structured probability distribution distance metric as the optimization objective enables the language model to effectively balance precision and recall.

## 4.2 LANGUAGE UNDERSTANDING

### 4.2.1 SETUP

**Pre-trained LLMs**  We adopt LLaMa-7B and LLaMa-13B (Touvron et al., 2023a) as the pre-trained LLMs. More results using LLaMa2-7B/13B (Touvron et al., 2023b) are deferred to Appendix B due to space limits.

**Continual Fine-tuning**  We perform continual fine-tuning on WikiText-103 using EMO and all baseline methods compared in Sec. 4.1. The corpus used for fine-tuning is substantially smaller (0.1B v.s. 1.4T tokens) than the corpus used for pre-training LLMs, therefore being much more efficient and resource-friendly. Using the same corpus for lightweight fine-tuning, our goal here is to explore the effect of different training objectives on downstream performance. For EMO, $\boldsymbol{E}$ is initialized from the pre-trained language modeling head and stay fixed during fine-tuning.

**Downstream Tasks**  We evaluate the fine-tuned models across an array of NLU tasks using in-context learning. Specifically, we use the following datasets: Tweet Emotion (Mohammad et al., 2018), TREC (Li & Roth, 2002; Hovy et al., 2001), SST-2 (Socher et al., 2013), Subj (Conneau & Kiela, 2018), Customer Review (Hu & Liu, 2004), Rotten Tomatoes (Pang & Lee, 2005), AG News (Zhang et al., 2015), and MMLU (Hendrycks et al., 2020). Following Wu et al. (2022; 2023), A pre-trained dense retriever is used to find the 8 most similar samples as the in-context demonstrations for all datasets except for MMLU, where a fixed 5-shot demonstrations are used following common practice. Prompt templates and statistics for each task can be found in Appendix A.5.

### 4.2.2 MAIN RESULTS

From Table. 3, we observe that continual fine-tuning using MLE often only marginally outperforms the pre-trained one and sometimes even hurts performance. The optimal performance of TaiLr and MixCE is obtained via grid search over the weighting coefficient from $\{0.9, 0.8, 0.1\}$. Notably, without any tunable hyperparameter, EMO yields the most significant gains across all tasks compared to existing methods upon both LLaMa-7B and LLaMa-13B, demonstrating the broader applicability of our method in terms of tasks, and model sizes.

Table 3: Downstream performance of LLaMa-7B/13B fine-tuned with different training objectives.

| Models | Methods | TE | SST-2 | TREC | Subj | CR | RT | AG | MMLU |
|---|---|---|---|---|---|---|---|---|---|
| LLaMa-7B | Pre-trained | 54.1 | 94.7 | 77.8 | 74.7 | 91.4 | 90.0 | 85.6 | 31.4 |
| | MLE | 53.5 | 94.8 | 79.0 | 74.5 | 92.0 | 91.8 | 85.5 | 31.9 |
| | TaiLr | 56.2 | 94.9 | 79.6 | 76.8 | 92.0 | 91.9 | 86.3 | 33.2 |
| | MixCE | 60.0 | 95.0 | 81.2 | 78.5 | 92.0 | 91.8 | 87.5 | 33.9 |
| | EMO | **65.6** | **95.2** | **83.4** | **79.2** | 92.0 | **92.1** | **89.4** | **34.8** |
| LLaMa-13B | Pre-trained | 58.5 | 95.6 | 81.2 | 77.4 | 91.2 | 91.0 | 84.5 | 44.5 |
| | MLE | 58.6 | 95.5 | 79.8 | 76.9 | 92.0 | 91.3 | 84.3 | 44.9 |
| | TaiLr | 61.9 | 95.5 | 81.0 | 78.5 | 92.3 | 91.4 | 85.6 | 45.9 |
| | MixCE | 65.7 | 95.6 | 82.8 | 80.6 | 92.0 | 91.3 | 85.9 | 46.7 |
| | EMO | **70.4** | **95.9** | **85.2** | **81.1** | **92.6** | **92.2** | **88.4** | **47.5** |

### 4.2.3 SCALING LAW OF EMO

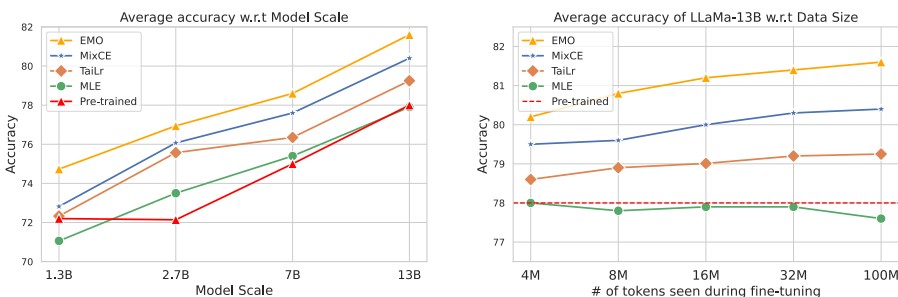

Figure 1: Scaling law of EMO with respect to model scale and data size.

**Model Scaling** To comprehensively quantify the effectiveness of EMO, we perform the previously described experiment upon OPT-1.3B/2.7B (Zhang et al., 2022) in addition to LLaMa-7B/13B and visualize the scaling curve of the task accuracy averaged over the collection of 8 datasets with respect to model scale in Fig.1 (left). While MLE fails to consistently improve over pre-trained models, TaiLr and MixCE both bring positive impacts when their weighting coefficients are carefully tuned. Notably, EMO shows steady improvements over other methods across all model scales.

**Data Scaling** We further examine how performance changes by varying data volumes during fine-tuning. We monitor the change of average accuracy using LLaMa-13B and display the results in Fig. 1 (right). MLE-tuned models exhibit certain declines in accuracy as fine-tuning progresses, which can be attributed to its theoretical deficiencies described in Sec. 2.2. TaiLr and MixCE moderately improve over MLE. EMO shows the most significant performance boost and even matches the performance of 100M-tokens-trained MixCE with merely 4M tokens. This highlights the potential of employing EMO in a post-training phase to refine the distribution of pre-trained LLMs for improved downstream performance in an effective and sample-efficient manner.

## 5 CONCLUSION

In this work, we introduce EMO, a novel approach for training auto-regressive language models by optimizing a differentiable upper bound of the earth mover distance between the model distribution and human text distribution. Experiments on open-ended text generation demonstrate that EMO consistently outperforms MLE and its robust baseline methods across diverse domains in terms of how human-like the texts generated from fine-tuned models are. Through a highly lightweight continual fine-tuning phase on unsupervised corpora, EMO can significantly enhance downstream performance compared to pre-trained models and exhibits commendable scaling properties regarding the amount of training data, rendering it favorable for general-purpose continual fine-tuning.

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

# A DISCUSSION AND EXPERIMENTAL DETAILS

## A.1 DISCUSSION OF BASELINE METHODS

In this paper, we mainly compare our method to baselines that attempt to improve MLE by optimizing distance measures beyond forward cross-entropy.

**TaiLr** Ji et al. (2022) proposes to leverage the total variation distance (TVD) as a more robust alternative to the forward cross-entropy for language generation model training. Specifically, they introduce a token-level factorization of the original TVD and optimize its upper bound in addition to the MLE loss. The training objective of TaiLr can be written as:

$$\mathcal{L}_{\text{TaiLr}} = -\frac{Q_\theta(x_t|x_{<t})}{\gamma + (1-\gamma)Q_\theta(x_t|x_{<t})} \log Q_\theta(x_t|x_{<t}) \tag{17}$$

From the form of Eq. 17 we can see that TaiLR only alleviates the recall-prioritization issue of MLE while still confronting the negative diversity ignorance and train-test mismatch problems.

**MixCE** Zhang et al. (2023) is another modification to MLE which incorporates the reverse cross-entropy into the training objective. Due to the one-hot encoded token-level data distribution, the author proposes an approximation and uses an interpolation coefficient to combine it with forward cross-entropy as follows:

$$\mathcal{L}_{\text{mixce}} = -(\gamma + (1-\gamma)Q_\theta(x_t|x_{<t})) \log Q_\theta(x_t|x_{<t}) \tag{18}$$

Both Eq. 17 and Eq. 18 can be regarded as the original $\mathcal{L}_{\text{MLE}}$ multiplied by a coefficient determined by a tunable hyper-parameter $\gamma$ and model's probability (confidence) of $x_t$. Though attractive in terms of their original formulation, TaiLr and MixCE both degenerate into certain regularized forms of MLE, hence demonstrating limited improvements.

## A.2 DIFFERENCES BETWEEN EMO AND REINFORCEMENT LEARNING FROM HUMAN FEEDBACK (RLHF)

Prevailing methodologies impart the desired behaviors into a base language model through meticulously crafted human preferences that represent the types of responses that humans find helpful. This stage, dubbed supervised fine-tuning (SFT), often happens after the initial unsupervised pre-training on a large text dataset. Although the STF models already exhibit good instruction-following capabilities, the common practice is to further align their behavior with human value, a procedure known as Reinforcement Learning from Human Feedback (RLHF) (Christiano et al., 2017; Ziegler et al., 2019; Bai et al., 2022).

The differences between EMO and RLHF manifest in multiple dimensions, including motivation, gradient, and application scenario. In the following, we discuss these points in detail.

- **Motivation**: The motivation behind EMO is to explore effective means to adapt a language model to a given human text dataset through the lens of earth-mover distance optimization. Evaluation is thus focused on quantifying how similar the model distribution $Q_\theta(\boldsymbol{x})$ is to human text distribution $P(\boldsymbol{x})$. In contrast, RLHF prioritizes steering the behavior of the language model based on the feedback provided by a specific reward model (PPO (Schulman et al., 2017)) or directly from existing human preference dataset (DPO (Rafailov et al., 2023)). The evaluation is often based on human-centric subjective metrics such as helpfulness and safety.

- **Gradient**: The per time step gradient of EMO is the combination of gradient of probability of **each token in the vocabulary**, weighted by their respective expected transport costs, i.e., $\sum_{i=1}^{|V|} \nabla_\theta Q_\theta(v_i)\mathbb{E}_{v_j \sim P}[C(v_i, v_j)]$. For PPO, the per time step gradient is the gradient of **current token**'s log probability, weighted by the reward $r(\boldsymbol{x}, \boldsymbol{y})$ and the deviation from a reference model $D_{KL}(Q_\theta(\boldsymbol{y}|\boldsymbol{x})||Q_{\text{ref}}(\boldsymbol{y}|\boldsymbol{x}))$, i.e., $\nabla_\theta \log Q_\theta(y_t|\boldsymbol{x}, \boldsymbol{y}_{<t})(r(\boldsymbol{x}, \boldsymbol{y}) - \log \frac{Q_\theta(y_t)}{Q_{\text{ref}}(y_t)})$. For DPO, the per time step gradient is the gradient of the **current token**(in the preferred or dispreferred response)'s log probability, weighted by the incorrectness value of an implicit reward model, i.e., $\nabla_\theta \log Q_\theta(y_t|\boldsymbol{x}, \boldsymbol{y}_{<t}) \cdot \sigma(\beta \log \frac{Q_\theta(y_l)}{Q_{\text{ref}}(y_l)} - \beta \log \frac{Q_\theta(y_w)}{Q_{\text{ref}}(y_w)})$.

- **Application Scenario**: As a general-purpose objective for auto-regressive language modeling, EMO is applicable to domain-specific fine-tuning/adaptation, instruction-tuning, and continual pre-training. Currently, the use of RLHF predominantly occurs in the alignment stage after supervised fine-tuning.

## A.3 DYNAMIC WEIGHTING

In situations where the language model is relatively weak (having high perplexity), DEMD may converge at a slow pace due to bounded gradient scaling induced by cosine-based transport cost. To overcome this potential issue, the final loss function in EMO is implemented as a dynamically weighted combination of MLE and DEMD:

$$\mathcal{L} = 0.5 * (\mathcal{L}_{\text{MLE}} + (\frac{\mathcal{L}_{\text{MLE}}}{\mathcal{L}_{\text{DEMD}}}).\text{detach}() * \mathcal{L}_{\text{DEMD}}) \tag{19}$$

## A.4 OPEN-ENDED GENERATION

| Datasets | WikiText2 | Wikitext103 | WebText | PTB | WritingPrompts | AG News |
|----------|-----------|-------------|---------|-----|----------------|---------|
| # of train samples | 36,700 | 1,800,000 | 20,000 | 4,210 | 10,000 | 112,000 |
| # of dev samples | 3,760 | 3,760 | 5,000 | 3,370 | 925 | 6,000 |
| # of test samples | 4,360 | 4.360 | 5,000 | 3,760 | 1,047 | 7,600 |
| prefix length | 20 | 20 | 20 | 5 | 35 | 10 |
| generation length | 80 | 80 | 80 | 25 | 80 | 30 |

Table 4: Length of the provided prefix and model generations for each dataset employed in the open-ended generation experiments.

We provide the detailed statistics and settings in the open-ended generation experiment in Table. 4. For WikiText2, Wikitext103, PTB, and AG News, we download the datasets from the HuggingFace Datasets hub. For WritingPrompts and WebText, we utilize the official split provided by Zhang et al. (2023).

### A.4.1 QUANTIFYING THE PRECISION-RECALL TRADEOFF

To have a quantitative understanding of how MLE is biased towards recall, we visualize the averaged token-level forward and reverse cross-entropy between $Q_\theta$ of a GPT-2 model fine-tuned with different objectives and that of a pre-trained GPT-Neo-1.3B (Black et al., 2021) model (which serves as a surrogate target distribution) in Fig. 2. TaiLr and MixCE demonstrate an improved balance between precision and recall, while our proposed EMO further outperforms these two methods with significant margins.

## A.5 PROMPT TEMPLATES FOR LANGUAGE UNDERSTANDING TASKS

The specific prompt templates used for each task in Sec. 4.2.3 are presented in Table.5. For MMLU, we follow the prompt design in Contributors (2023) [1]. In implementation, each test input is prepended with 8 demonstrations that are retrieved using a pre-trained dense retriever based on semantic similarity. One exception is MMLU, where we adopt a fixed 5-shot demonstration following previous works. We compute the perplexity for the constructed prompt corresponding to each candidate answer and choose the one with the smallest perplexity as the final prediction. Evaluations are implemented based on the OpenICL (Wu et al., 2023) library.

## B ADDITIONAL RESULTS ON LANGUAGE UNDERSTANDING

The additional results of LLaMa2-7B, LLaMa2-13B , and OPT-2.7B on downstream natural language understanding tasks evaluated in Sec. 4.2.3 are summarized in Table. 6, Table. 7, and Table. 8 respectively. LLMs fine-tuned with our proposed method display notable improvements over MLE and strong baselines in most tasks.

---

[1] `https://github.com/open-compass/opencompass`.

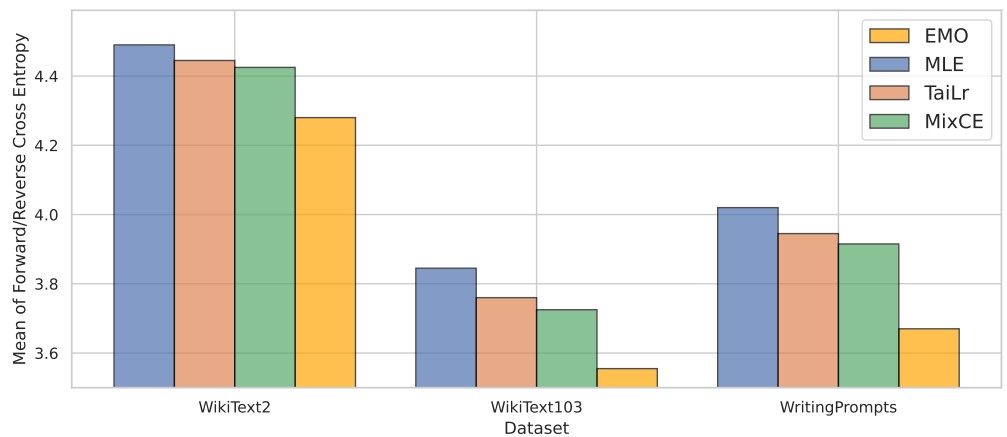

Figure 2: The average of token-level forward and reverse cross-entropy between distribution $Q_\theta$ of GPT-2 fine-tuned with different objectives and that of GPT-Neo-1.3B on the validation set of three different datasets. The lower the value, the better the learned $Q_\theta$ balance precision and recall.

| Tasks | Prompts | Label |
|---|---|---|
| SST-2 | Review: "<X>" It is positive.
Review: "<X>" It is negative. | Positive
Negative |
| Tweet Emotion | Tweet: "<X>" It is anger.
Tweet: "textlessX>" It is joy.
Tweet: "<X>" It is optimism.
Tweet: "<X>" It is sadness. | Anger
Joy
Optimism
Sadness |
| TREC | Question: "<X>" It is about abbreviation.
Question: "<X>" It is about entity.
Question: "<X>" It is about description and abstract concept.
Question: "<X>" It is about human being.
Question: "<X>" It is about location.
Question: "<X>" It is about numerical value. | Abbreviation
Entity
Description and abstract concept
Human being
Location
Numerical value |
| Subj | "<X>" It is objective.
"<X>" It is subjective. | Objective
Subjective |
| CR | Review: "<X>" It is positive.
Review: "<X>" It is negative. | Positive
Negative |
| Rotten Tomatoes | Review: "<X>" It is positive.
Review: "<X>" It is negative. | Positive
Negative |
| AG News | "<X>" It is about world.
"<X>" It is about sports.
"<X>" It is about business.
"<X>" It is about science and technology. | World
Sport
Business
Science and Technology |

Table 5: Prompt templates for natural language understanding tasks used in Sec .4.2.3. <X> is a placeholder that indicates the real input context/question.

| Methods | TE | SST-2 | TREC | Subj | CR | RT | AG | MMLU |
|---|---|---|---|---|---|---|---|---|
| Pre-trained | 56.7 | 95.4 | 74.6 | 76.2 | **93.6** | 91.6 | 86.2 | 43.6 |
| MLE | 58.4 | 95.8 | 72.4 | 74.6 | 93.4 | 92.1 | 86.0 | 43.3 |
| TaiLr | 62.4 | 95.9 | 74.6 | 81.0 | 92.8 | 92.5 | 87.4 | 44.4 |
| MixCE | 66.5 | **95.9** | 77.6 | 82.5 | 93.1 | 92.1 | 88.0 | 45.1 |
| EMO | **69.0** | 95.8 | **78.0** | **83.1** | 93.4 | **92.8** | **88.1** | **45.5** |

Table 6: Downstream task performance of LLaMa2-7B fine-tuned with different training objectives on WikiText-103.

| Methods | TE | SST-2 | TREC | Subj | CR | RT | AG | MMLU |
|---|---|---|---|---|---|---|---|---|
| Pre-trained | 60.4 | 95.5 | 80.4 | 81.7 | 91.2 | 90.1 | 86.5 | 54.7 |
| MLE | 60.9 | 95.8 | 81.0 | 81.8 | 90.9 | 89.6 | 86.2 | 54.3 |
| TaiLr | 61.7 | 96.2 | 80.4 | 81.4 | **91.5** | 90.5 | 87.2 | 54.5 |
| MixCE | 64.4 | 95.7 | 83.6 | 84.7 | 91.2 | 90.4 | 87.4 | 55.0 |
| EMO | **70.7** | **96.2** | **85.8** | **89.9** | 91.2 | **91.7** | **89.6** | **55.2** |

Table 7: Downstream task performance of LLaMa2-13B fine-tuned with different training objectives on WikiText-103.

| Methods | TE | SST-2 | TREC | Subj | CR | RT | AG | MMLU |
|---|---|---|---|---|---|---|---|---|
| Pre-trained | 65.7 | 92.9 | 68.6 | 85.5 | 92.5 | 87.6 | 84.3 | 24.4 |
| MLE | 67.3 | 88.7 | 66.8 | 83.4 | 91.5 | 81.3 | 84.4 | 24.9 |
| TaiLr | 67.1 | 92.9 | 70.4 | 88.2 | 92.6 | 87.1 | 84.0 | 24.7 |
| MixCE | 66.8 | 93.1 | 70.6 | **88.8** | 92.8 | **87.4** | 84.1 | 25.0 |
| EMO | **68.6** | **93.7** | **73.6** | 87.6 | **92.8** | 86.0 | **87.6** | **25.6** |

Table 8: Downstream task performance of OPT-2.7B fine-tuned with different training objectives on WikiText-103.

## C  INSTRUCTION-TUNING

The effectiveness of Large Language Models (LLMs) heavily relies on their capacity to comprehend precise instructions. These generative language models undergo training using extensive raw web data and are further refined through a meticulous selection of instruction data, albeit in a relatively limited amount. The process of fine-tuning with instructions plays a pivotal role in harnessing the potential of LLMs. Consequently, the utility of such models is predominantly shaped by our proficiency in maximizing their performance using compact instruction datasets.

To this end, we also apply EMO to the instruction-tuning stage of LLaMa-7B/13B using the Alpaca-GPT4 dataset (Peng et al., 2023). In addition, we perform experiments using a more advanced instruction-tuning dataset, i.e., Recycled Evol-Instruct-70K proposed by Li et al. (2023b), as well as OpenPlatypus (Lee et al., 2023), a curated dataset derived from 11 open-source datasets, primarily focusing on enhancing LLMs' STEM and logic proficiency. We follow the standard training recipe (3 training epochs, 128 global batch size, 2e-5/1e-5 learning rate for 7B/13B models respectively) adopted in the original Stanford Alpaca repository [2]. Afterwards, we assess the instruction-adherence efficacy of the resulting instruction-tuned models by incorporating the following recent LLM-based evaluation methods:

- AlpacaEval (Li et al., 2023c), which is an LLM-based automatic evaluation that is fast, cheap, and replicable. We adopt GPT-4 as the evaluator and report the win rate against responses generated by text-davinci-003.

- Auto-J (Li et al., 2023a), which is a 13B parameter open-source generative judge that can effectively evaluate different LLMs on how they align to human preference. We report the win and tie counts of models fine-tuning using MLE and EMO.

- PandaLM (Wang et al., 2023), a 7B parameter instruction-tuned LLM that aims to provide reproducible and automated comparisons between different large language models.

We empirically found that both Auto-J and PandaLM fail to distinguish the differences between lengthy responses. Therefore, we only apply them to evaluate models trained on Alpaca-GPT4, in which the reference responses are much shorter. As indicated in Table. C, EMO-tuned LLaMa attains superior success rates in comparison to MLE-tuned counterparts across various model sizes. The average response length (measured by the number of tokens following tokenization) for MLE and EMO are 233 and 226, respectively. This shows that EMO-tuned models are able to produce higher-quality responses without relying on GPT-4's bias towards length and verbosity. In pairwise

---

[2]`https://github.com/tatsu-lab/stanford_alpaca`.

| | Training Objective | AlpacaEval Win Rate(%) |
|---|---|---|
| LLaMa-7B | MLE | 59.3 |
| | EMO | 68.4 |
| LLaMa-13B | MLE | 70.3 |
| | EMO | 74.2 |
| LLaMa2-7B | MLE | 59.3 |
| | EMO | 70.3 |
| LLaMa2-13B | MLE | 67.3 |
| | EMO | 79.1 |

Table 9: AlpacaEval win rate of LLaMa-7B/13B and LLaMa2-7B/13B fine-tuned with MLE and EMO on **Alpaca-GPT4** against text-davinci-003 on 805 test instructions.

| | Training Objective | AlpacaEval Win Rate(%) |
|---|---|---|
| Evol-Instruct-70K | MLE | 76.2 |
| | EMO | 78.8 |
| OpenPlatypus | MLE | 58.9 |
| | EMO | 63.0 |

Table 10: AlpacaEval win rate of LLaMa2-7B fine-tuned with MLE and EMO on **Recycled Evol-Instruct-70K** and **OpenPlatypus** against text-davinci-003 on 805 test instructions.

evaluation performed by Auto-J and PandaLM (Fig. 3, 4, 5, 6), EMO-tuned models also achieve higher win rates over MLE, further verifying the superiority of EMO when applied to instruction-tuning. Due to higher model capacity and more comprehensive rationale for decision making, Auto-J is more capable of differentiating the quality between different responses, while PandaLM consistently produces more "tie".

In light of the efficiency and commendable performance of Auto-J, we further adopt it to compare the respones produced by MLE/EMO-tuned LLaMa2-7B against the publically available respones from a wide variety of instuction-following models on the AlpacaEval leaderboard. The win rates are shown in the table below.

| Competetor Model | Evol-Instruct-70K | | OpenPlaytpus | |
|---|---|---|---|---|
| | MLE | EMO | MLE | EMO |
| Davinci-003 | 91% | 93% | 77% | 78% |
| Baize-v2-7B | 82% | 86% | 57% | 59% |
| LLaMa2-7B-Chat | 63% | 65% | 37% | 39% |
| Vicuna-7B-v1.3 | 76% | 80% | 48% | 52% |
| Zephyr-7B-alpha | 70% | 72% | 40% | 44% |

Table 11: Auto-J judged win rates of MLE/EMO-tuned LLaMa2-7B on Evol-Instruct-70K and OpenPlatypus against publically available responses from various close-sourced and 7B-sized open-source models.

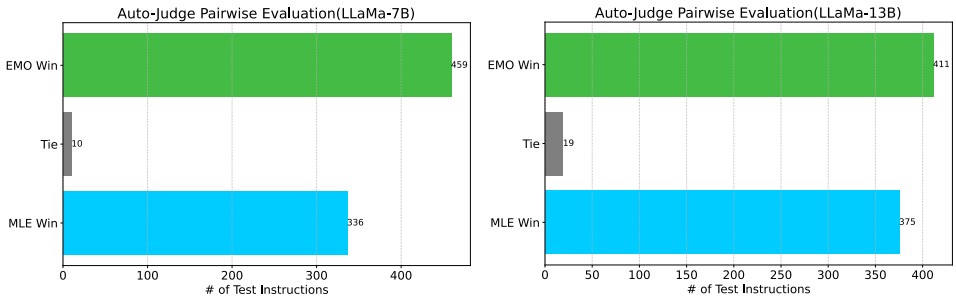

Figure 3: Auto-J pairwise response comparison results of LLaMa-7B/13B fine-tuned with MLE and EMO on 805 test instructions from AlpacaEval.

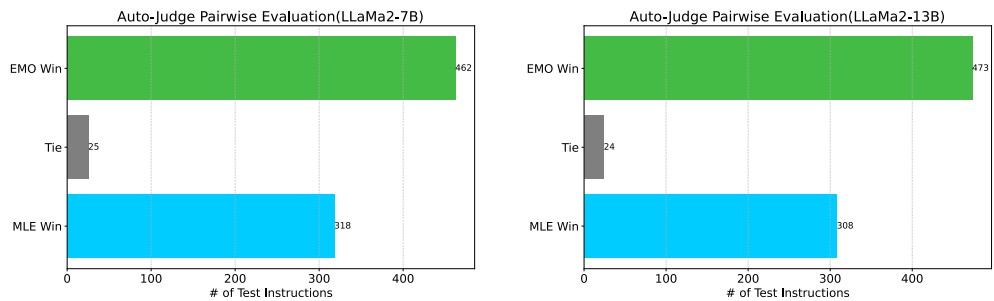

Figure 4: Auto-J pairwise response comparison results of LLaMa2-7B/13B fine-tuned with MLE and EMO on 805 test instructions from AlpacaEval.

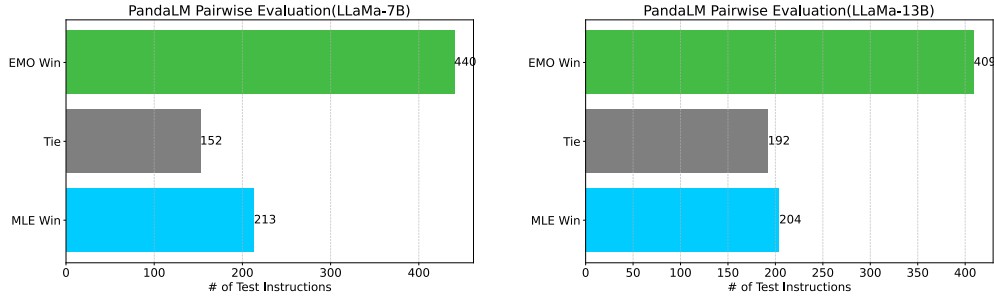

Figure 5: PandaLM pairwise response comparison results of LLaMa-7B/13B fine-tuned with MLE and EMO on 805 test instructions from AlpacaEval.

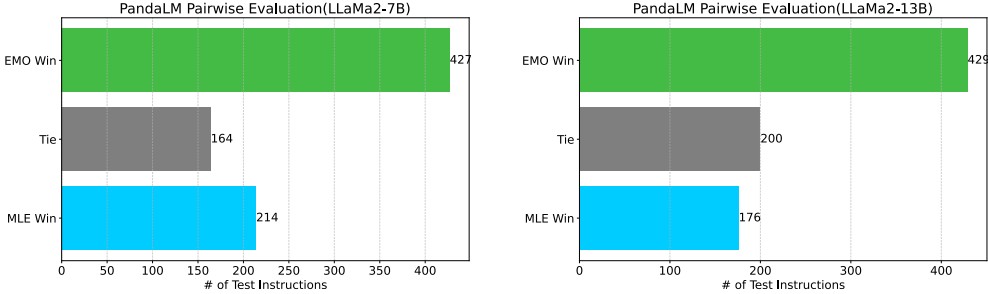

Figure 6: PandaLM pairwise response comparison results of LLaMa2-7B/13B fine-tuned with MLE and EMO on 805 test instructions from AlpacaEval.

