# OpenReview forum: "EMO: EARTH MOVER DISTANCE OPTIMIZATION FOR AUTO-REGRESSIVE LANGUAGE MODELING"
_ICLR.cc/2024/Conference — ICLR 2024 poster_

### Official Review · Reviewer_XE26 · 2023-10-19

**Soundness:** 1 poor
**Presentation:** 3 good
**Contribution:** 2 fair
**Rating:** 3
**Confidence:** 4

**Summary:**

This paper presents a training objective called Earth Mover Distance Optimization (EMO) for autoregressive language models. In contrast to maximum likelihood estimation, which solely enhances the likelihood of the ground-truth token, EMO takes into account the embedding similarity between predicted tokens and the ground-truth token, promoting predictions of tokens with high embedding similarity. Experimental results demonstrate that EMO generates superior outputs compared to MLE when employing unbiased sampling.

**Strengths:**

This paper is easy to follow. The presentation is mostly clear.

**Weaknesses:**

1. **There are significant concerns regarding the evaluation methodology in this paper. Comparing MLE and EMO based on outputs generated by unbiased sampling is unfair, as the two objectives result in considerably different model distributions.** The expected MLE loss is minimized when the model distribution Q matches the data distribution P. As mentioned in the paper, the expected EMO loss is $E_{v_i ~ Q}[\sum_{j=1}^{|V|}P(v_j)C(v_i, v_j)]$, which is minimized when the model distribution Q is a one-hot distribution that only outputs the token i that maximizes $\sum_{j=1}^{|V|}P(v_j)C(v_i, v_j)$. Consequently, models trained using EMO will predict a much sharper distribution, leading to higher-quality but lower-diversity outputs when sampling from EMO. The evaluation methodology in this paper, i.e., comparing the sampling results of EMO and MLE, is therefore like comparing the output quality of greedy decoding and unbiased sampling, which is inherently unfair. The fair way is to compare their decoding results of greedy/beam search, necessitating further experiments to establish the effectiveness of EMO.

2. Based on the above analysis, the authors' motivation appears to be flawed. Under ideal circumstances, MLE trains the model to conform to the data distribution. In contrast, EMO results in a one-hot distribution, which is recall-prioritization and negative diversity ignorance.

3. The proposed EMO loss is very similar to the word embedding-based loss [1]. The only difference lies the choices of distance mertics (Cosine distance in EMO, Euclidean distance in [1]).

[1] https://arxiv.org/pdf/1807.11219.pdf

**Questions:**

None

---

> ### Author Response · Authors · 2023-11-16
>
> Dear reviewer, we sincerely appreciate your time reading the paper. Here we do our best to address your concerns.
> - **MLE/EMO-induced distribution**: For per-token loss, both MLE and EMO are minimized **theoretically** when the model distribution $Q_{\theta}$ is one-hot. However, in practical training which involves iterative stochastic gradient descent on human text data, both MLE and EMO optimize model distribution to become more human-like. Moreover, as stated in Section 4.1.1 Training Details, the final loss of EMO is the additive combination of MLE and DEMD. The difference between the final distributions induced by MLE and EMO emerge as the aggregated effect of per-token gradient update throughout training, where EMO explicit penalizes low-quality tokens and takes more account into plausible alternatives. We demonstrate this quantitatively and qualitatively
> - **Quantitative difference**: We show the differences between MLE and EMO induced distributions by measuring the forward cross-entropy $\text{CE}^{\text{forward}}=-\sum{P\log{Q_{\theta}}}$, reverse cross-entropy $\text{CE}^{\text{reverse}}=-\sum{Q_{\theta}\log{P}}$, and their harmonic mean against a reference distribution $P$ provided by a larger language model GPT-Neo. The GPT-2 results of $Q_{\theta}^{\text{MLE}}$, $Q_{\theta}^{\text{EMO}}$, and a smoothed one-hot distribution are shown in the table below. As can be seen, both $Q_{\theta}^{\text{EMO}}$ and $Q_{\theta}^{\text{MLE}}$ deviate far from one-hot and $Q_{\theta}^{\text{EMO}}$ is **more similar to reference distribution** by better harmonizing recall and precision.
> |  |  | WikiText-2 |  |  | WritingPrompt |  |
> |:---:|:---:|:---:|:---:|:---:|:---:|:---:|
> |  | $\text{CE}^{\text{reverse}}$↓ | $\text{CE}^{\text{forward}}$↓ | Harmonic Mean↓ | $\text{CE}^{\text{reverse}}$↓ | $\text{CE}^{\text{forward}}$↓ | Harmonic Mean↓ |
> | $Q^{\text{one-hot}}$ | 3.22 | 16.83 | 5.15 | 3.32 | 17.74 | 5.33 |
> | $Q_{\theta}^{\text{MLE}}$ | 4.50 | 4.33 | 4.22 | 4.25 | 3.92 | 3.99 |
> | $Q_{\theta}^{\text{EMO}}$ | 3.94 | 4.45 | 3.97 | 3.54 | 4.01 | 3.66 |
>
>     We also report the Distinct N-gram of the sampled outputs from $Q_{\theta}^{\text{MLE}}$, $Q_{\theta}^{\text{EMO}}$, and reference text. As can be seen,  $Q_{\theta}^{\text{EMO}}$ is also more similar to human text distribution in various aspects of the sampled outputs.
> |  |  |  | WikiText2 |  |  |  |  |  | WritingPrompt |  |  |
> |---|:---:|:---:|:---:|:---:|:---:|---|:---:|:---:|:---:|---|:---:|
> |  |  |  | Dist$_{n}$ |  | MAUVE |  \|  |  |  | Dist$_{n}$ | | MAUVE |
> |  | n=1 | n=2 | n=3 | n=4 | - | \| | n=1 | n=2 | n=3 | n=4 | - |
> | $Q_{\theta}^{\text{MLE}}$ | 0.748 | 0.971 | 0.994 | 0.998 | 77.5 | \| | 0.806 | 0.983 | 0.995 | 0.997 | 83.6 |
> | $Q_{\theta}^{\text{EMO}}$ | 0.698 | 0.955 | 0.990 | 0.997 | 87.5 | \| | 0.762 | 0.972 | 0.993 | 0.996 | 87.4 |
> | Human | 0.687 | 0.958 | 0.988 | 0.996 | 100 | \| | 0.764 | 0.974 | 0.993 | 0.997 | 100 |
> - **Qualitative difference**: Here we show some predicted next token probability from $Q_{\theta}^{\text{MLE}}$ and $Q_{\theta}^{\text{EMO}}$. We show the top 5 predicted tokens and their probabilities.
>     - Prefix: Donald Trump has multiple occupations, such as
>     - $Q_{\theta}^{\text{MLE}}$: the(0.393), a(0.325), construction(0.100), executive(0.093), president(0.088),...
>     - $Q_{\theta}^{\text{EMO}}$: a(0.366), the(0.353), president(0.100), executive(0.091), construction(0.091),...
>     - Prefix: Elephant is the largest living
>     - $Q_{\theta}^{\text{MLE}}$: animal(0.275),creature(0.231),elephant(0.215),mammal(0.197),ant(0.079),...
>     - $Q_{\theta}^{\text{EMO}}$: animal(0.268),creature(0.257),mammal(0.216),elephant(0.196),rept(0.062),...
>
> - **Greedy/Beam search**: The results of greedy/beam search of $Q_{\theta}^{\text{MLE}}$ and $Q_{\theta}^{\text{EMO}}$ on WikiText-2 test set are shown in the table below. **More greedy decoding results of LLaMa2 are presented in our General Response**.
> |  | Method | MAUVE | BLEU | ROUGE-1 |
> |---|:---:|:---:|:---:|:---:|
> | Greedy Search | $Q_{\theta}^{\text{MLE}}$ | 7.10 | 21.64 | 38.8 |
> |  | $Q_{\theta}^{\text{EMO}}$ | 11.0 | 22.27 | 38.0 |
> | Beam Search | $Q_{\theta}^{\text{MLE}}$ | 9.93 | 21.67 | 37.0 |
> |  | $Q_{\theta}^{\text{EMO}}$ | 13.14 | 22.37 | 37.2 |
>
> - **Difference between EMO and word embedding-based loss**: We would like to modestly point out that the word embedding-based loss described in the paper mentioned by the reviewer is indeed a weighted variant of MLE. In their method, the gradient of the **ground-truth token's probability** is weighted by the sum of its Euclidean distance to all tokens in the target vocabulary, .i.e, $\nabla Q_{\theta}(y_i)\sum_k d(E(v_k),E(v_i))$. In contrast, EMO involves the sum of gradients of **all tokens' probability**, weighted by their respective expected transport cost. We will cite this paper in the final version.
>
> Reference:
>
> [1] Meister et. al: On the Efficacy of Sampling Adapters. ACL 2023

---

> ### Comment · Reviewer_XE26 · 2023-11-17
>
> Thanks for your reply.
>
> 1. I appreciate the clarification. I didn't notice that the final loss also includes MLE, so it turns out that the distribution of EMO is not as sharp as I expected. However, I disagree with the author's claim that "both MLE and EMO optimize model distribution to become more human-like." Based on my analysis, the expected EMO loss is minimized when the model distribution Q is a one-hot distribution that only outputs the token i that maximizes $\sum_{j=1}^{|V|}P(v_j)C(v_i, v_j)$. Consequently, the perplexity of EMO is higher than that of MLE, indicating a less human-like distribution. If DEMD were used solely as the loss function, I believe the resulting model distribution would be sharper and less human-like, leading to a significantly higher perplexity. A similar phenomenon can be observed in language GANs [1].
>
> 2. I maintain my point that the paper's motivation is flawed, and I suggest the author consider presenting the story of EMO from a different perspective.
>
> 3. I acknowledge my previous misunderstanding of the word embedding-based loss and appreciate the clarification provided.
>
> I have a few questions:
>
> 1. Could you explain the concept of Q^{one-hot}?
>
> 2. In the WikiText-2 test set results for greedy/beam search, why are the MAUVE scores substantially lower than those obtained through unbiased sampling, as shown in Table 1?
>
> [1] https://arxiv.org/pdf/1811.02549.pdf

---

> ### Author Response · Authors · 2023-11-17
>
> Thanks for reading our responses. Regarding your questions, our answers are as follows:
> - The concept of $Q^{\text{one-hot}}$ refers to the one-hot distribution constructed based on the ground-truth next token.
>  The various divergence measures of $Q^{\text{one-hot}}$ we've reported in our response serve as a reference to indicate how far are $Q_{\theta}^{\text{EMO}}$) from such one-hot distribution. Compared to $Q_{\theta}^{\text{MLE}}$, $Q_{\theta}^{\text{EMO}}$ is **closer** to the reference distribution $P$ as measured by the harmonic mean of forward/reverse cross-entropy.
> - This is because MAUVE score measures how similar the model distribution $Q_{\theta}$ is to the data distribution $P$. Concretely, MAUVE computes the area under an interpolated KL divergence curve. **It rewards model distribution that balances recall($Q_{\theta}$ has sufficient cover of $P$'s mode) and precision($Q_{\theta}$ does not overestimates regions where $P$ is low)**. Two identical distributions will have a MAUVE score of 100. **Both greedy and beam search are maximization-based decoding strategies that discard all possible outputs except the most possible one**. This is equivalent to a high-precision but significantly low-recall output distribution. Therefore, the MAUVE scores of greedy and beam search are usually much smaller than sampling-based methods, e.g., unbiased/top-p/top-k/typical sampling. The same phenomenon was also observed in the original MAUVE paper as well as a prior work[1]'s Table 1.
>
> In the following we would like to explicate more about your comments:
> - We would like to kindly clarify that, in the context of auto-regressive language modeling(where we have a collection of text corpus for training), the mathematically lowest loss at each time step is achieved when the ground-truth next token's probability under $Q_{\theta}$ is 1, for both MLE and DEMD: (1) for MLE, -log1=0, and (2) for DEMD, 1-1*1*cos(0)=0. Both MLE and EMO optimize model towards such solution, but through different trajectories, i.e., gradient signal. The gradient signal of EMO informs more about "bad" tokens and alternative "good" tokens.
> - "higher perplexity": MLE has lower perplexity is expected because MLE's objective is to minimize perplexity($e^{-\log{Q_{\theta}(x)}}$), which is the so-called recall-prioritization[2]. EMO has higher perplexity because it cares more about diversity in the range of other "good" tokens.
> - "less human-like distribution": According to an exhaustive array of quantitative metrics we've reported, i.e., MAUVE, Harmonic mean of forward/reverse cross-entropy, Distinct N-gram, and the alignment experiments upon LLaMa2, the EMO-induced distribution is more human-like. We would much appreciate if the reviewer could elaborate further why EMO is less human-like.
>
> Please let us know you have further questions and we will try our best to solve your issues.
>
> References:
>
> [1] Su et.al: A Contrastive Framework for Neural Text Generation. NeurIPS 2022.
>
> [2] Meister et. al: On the Efficacy of Sampling Adapters. ACL 2023

---

> > ### Comment · Reviewer_XE26 · 2023-11-19
> >
> > Thanks for your clarification. When analyzing the model distribution induced by loss, we need to analyze the expected loss with respect to the data distribution, rather than the loss of individual samples. The expected MLE loss, $-\sum P \log Q$, is minimized when $P=Q$, which is a well known property of MLE. On the other hand, the expected EMO loss, $E_{v_i ~ Q}[\sum_{j=1}^{|V|}P(v_j)C(v_i, v_j)]$, is minimized when the model distribution Q is one-hot that only outputs the token i that maximizes $\sum_{j=1}^{|V|}P(v_j)C(v_i, v_j)$. Consequently, I believe that using EMO as the sole loss function would result in a sharper, less human-like model distribution, leading to a significantly higher perplexity (the perplexity is also minimized when $P=Q$).
> >
> > The current implementation, which combines MLE and EMO to train the model, seems to mitigate this issue to some extent, allowing for an improvement in quality without severely compromising diversity. As long as the authors enhance the presentation of the results by simultaneously displaying the model's quality and diversity, as well as their trade-off (e.g., through temperature sampling), the experimentation will be satisfactory.
> >
> > However, I still maintain a negative score because the motivation is flawed. The authors argue that MLE is recall-prioritization and negative diversity ignorance, a claim with which I disagree. Based on the analysis above, using EMO as the loss function would theoretically induce a one-hot distribution, which is recall-prioritization and negative diversity ignorance. Therefore, I am against the publication of this paper, as it will be misleading to the community.

---

> > > ### Author Response · Authors · 2023-11-22
> > >
> > > Dear reviewer, we greatly appreciate your response. We try our best to address your concerns as follows:
> > >
> > > - The setting of language modeling task:
> > >
> > >     - Under an idealized setting where we have unlimited high-quality text data, sufficient model capacity, and perfect optimizer, learning with MLE will lead to $Q_{\theta}$ as close to $P$ as we like, i.e., the optimal solution of $KL(P||Q_{\theta})$ is for $Q_{\theta}$ to converge to $P$. However, in practical language modeling, we only have access to finite and potentially noisy data. Moreover, the token-level target distribution at each time step is one-hot. The recall-prioritization and negative diversity ignorance issues of MLE arise under the inescapable constraints of such circumstances.
> > >
> > > - DEMD under ideal setting:
> > >
> > >     - This work builds upon the constraints of MLE in real-world scenarios for language modeling. In the context of auto-regressive language modeling, the optimal solution of $Q_{\theta}$ for MLE and DEMD both converge to the target data distribution(the one-hot distribution $P$). If the reviewer persists in the belief that we should explore the optimal scenario wherein a dense target data distribution $P_{\text{dense}}$ is present at each time step, we also present a variant of DEMD tailored for optimization in such a setting: $DEMD_{\text{dense}}=|Q_{\theta}-P_{\text{dense}}|^T CP_{\text{dense}}$. The optimization process involves minimizing the difference between the DEMD of $Q_{\theta}$ and $P_{\text{dense}}$, with the optimal solution for $Q_{\theta}$ being $P_{\text{dense}}$. The derivation process will be incorporated into the manuscript.
> > >
> > > - "The authors argue that MLE is recall-prioritization and negative diversity ignorance, a claim with which I disagree":
> > >
> > >     - We would like to kindly clarify that we are not the first to establish these two notions. Specifically, the notion of **recall-prioritization** was posed and targeted as the main research subject of several preceding endeavors[1,2,3], either by exploring the effects of various post-hoc truncation-based sampling strategies or designing alternative divergence measures for optimization during training. The notion of **negative diversity ignorance** was also discussed and studied by literature in the general NLG community[4,5,6], involving methodologies mainly built upon reference-based likelihood calibration in traditional supervised learning settings, e.g., abstractive summarization.
> > >
> > >         Similar to prior works mentioned above that incorporate various modifications in the training objective into MLE, we also incorporate DEMD into MLE for joint optimization. The reason is that the oracle transport cost $C$ is hard to obtain and our implementation of it is just one reasonable heuristic. Future work may involve the exploration of more effective implementations. Nonetheless, we have quantitatively shown that EMO delivers better harmonization of recall and precision, higher diversity in the range of plausible tokens, which ultimately lead to more human-like distribution as indicated by higher MAUVE score, Distinct N-gram, and distributional measures(harmonic mean of forward/reverse cross-entropy).
> > >
> > >
> > >
> > > Please let us know if you have further questions.
> > >
> > > References:
> > >
> > > [1] Meister et. al: [On the Efficacy of Sampling Adapters](https://aclanthology.org/2023.acl-long.80/). ACL 2023.
> > >
> > > [2] Zhang et. al: [MIXCE: Training Autoregressive Language Models by Mixing Forward and Reverse Cross-Entropies](https://aclanthology.org/2023.acl-long.502.pdf). ACL 2023.
> > >
> > > [3] Ji et. al: [Tailoring Language Generation Models under Total Variation Distance](https://openreview.net/pdf?id=VELL0PlWfc). ICLR 2023.
> > >
> > > [4] Li et. al: [Data-Dependent Gaussian Prior Objective for Language Generation](https://openreview.net/pdf?id=S1efxTVYDr). ICLR 2020.
> > >
> > > [5] Liu et. al: [BRIO: Bringing Order to Abstractive Summarization](https://aclanthology.org/2022.acl-long.207). ACL 2022.
> > >
> > > [6] Zhao et. al: [Calibrating Sequence Likelihood Improves Conditional Language Generation](https://arxiv.org/abs/2210.00045). ICLR 2023.

---

> > > > ### Comment · Reviewer_XE26 · 2023-11-23
> > > >
> > > > Thanks for your reply.
> > > >
> > > > I don't agree that we can justify focusing solely on one-hot data distribution due to the limited size of training set. One of the most important characteristics of Deep Neural Networks is their exceptional generalization ability, allowing them to perform well on unseen data after being trained on a limited set of examples. Despite the limited amount of training data and their statistical distribution being primarily one-hot, we still should analyze the theoretical properties of model based on the real data distribution, as the model possesses the ability to generalize to real distribution.
> > > >
> > > > A simple counterexample is that if we only consider one-hot data distribution, the negative log-likelihood loss $-\log Q(x_t|x_{<t})$ and negative probability loss $-Q(x_t|x_{<t})$ are equivalent, both making the model converge to the data distribution. However, on the wikitext-103 dataset, fine-tuning a MLE-trained model with negative probability loss for just one epoch increases the perplexity from 27 to 2000, indicating a significant deviation of the model distribution from the data distribution. Additionally, the average prediction entropy of the model reduces from 3.14 to 0.46, suggesting a much sharper model distribution. This phenomenon can be well-explained if we consider the expected loss under the real data distribution. The model distribution that minimizes the negative probability loss is one-hot, which outputs $argmax_{x_t}P(x_t|x_{<t})$, so training with this kind of loss would severely damage the model's diversity.
> > > >
> > > > Regarding recall-prioritization and negative diversity ignorance, I checked the provided references and find that some of them mention these concepts [1,2] but most of these references are irrelevant. Thus, I don't think that these two properties of MLE are widely recognized.
> > > >
> > > > [1] Meister et. al: On the Efficacy of Sampling Adapters. ACL 2023.
> > > > [2] Li et. al: Data-Dependent Gaussian Prior Objective for Language Generation. ICLR 2020.

---

### Official Review · Reviewer_JZqx · 2023-10-31

**Soundness:** 4 excellent
**Presentation:** 4 excellent
**Contribution:** 2 fair
**Rating:** 6
**Confidence:** 4

**Summary:**

This paper introduces a novel loss function, EMO. The main idea behind emo is to soften the loss function (vanilla cross entropy) based on the cosine similarity of pretrained lm head embeddings. EMO demonstrates consistently better than MLE across models, tasks, scales, and metrics.

**Strengths:**

1. This paper is well-organized and well-written, so it is generally easy to follow.
2. The intuition of the idea makes sense, and it is good that the complex earth mover distance loss can be reformulated into the cosine similarity dot probability.
3. The author(s) conducted extensive and detailed experiments, encompassing different model sizes, data scales, and evaluation metrics (which is very important). The thorough and detailed experiments show the advantages of EMO, adding to the soundness of the paper.

**Weaknesses:**

I have two concerns here.

1) Originality of the method.
In my view, the final loss function is very similar to the d2gpo loss, the authors did cite the d2gpo paper, but they ignored the methodology comparison and they should add d2gpo as a baseline.

2) Why Cosine Similarity?
Using cosine similarity is a choice, but may not be the best choice. The cosine similarity relies on the pre-trained llm head embedding, which makes it not unbiased. And through the 3.3 section, the gradient of the proposed EMO is very similar to the REINFORCE with cosine similarity as the reward, so maybe RLHF/RLAIF will be a better reward model?

**Questions:**

pls see weakness and the following questions:

3) why EMO can make ppl better? ppl is directly related to MLE, in other words, there does not exist a training-test mismatch problem. Is that because the evaluation is conducted over the sampled sentences instead of the test set. If so, please also report the ppl in another table.

4) please also compare with label smoothing, which is also a very useful loss function in the era of pre-llm.

5) what does the accuracy mean in sec 4.2.3?

---

> ### Author Response · Authors · 2023-11-16
>
> Dear reviewer, we sincerely appreciate your time reading the paper. Here we do our best endeavor to address your concerns:
> - **Choice of Baselines**: Our primary rationale for selecting baselines is to compare methods that optimize different **probability divergence measures** between model distribution and data distribution. Among our baselines, MLE optimizes the **forward cross-entropy**, TaiLr focuses on the **total variation distance** and MixCE mixes **forward and reverse cross-entropy**. D2GPo did not propose to optimize new divergence measures but instead focused on a **new target distribution**. Moreover, we didn't find the publicly available official implementation of D2GPo. Therefore, we didn't include D2GPo as our baseline in our submitted draft. Our reimplementation of D2GPo shows similar performance to MLE.
> - **Choice of Cosine Distance**: The reasons for using cosine distance between llm head embedding are three-fold:
>   - The general quadratic form of DEMD using a $|V|\times|V|$ transport cost matrix $C$ is $O(|V|^2)$. By utilizing the cosine distance, we can decrease the computation complexity from $O(|V|^2)$ to $O(|V|H)$(Eq. 13->Eq. 14).
>   - The output logits of LM are computed based on the dot-product between hidden states and LLM head embedding. The computation of cosine distance is also based on dot-product between normalized llm head embedding, exploiting the information stored in llm head. In fact, the transport cost matrix can also be computed based on the lm head of the model undergoing fine-tuning, which gets updated to incorporate the distribution shift from the pre-training corpus to task-specific training data. We term this variant of EMO as $\text{EMO}_{\text{adaptive}}$and conduct experiments to verify this and the results of language modeling are shown in the table below:
> |  | WikiText-2 | WritingPrompts |
> |:---:|:---:|:---:|
> | MLE | 77.5 | 83.6 |
> | EMO | 87.5 | 87.4 |
> | EMO$_{\text{adaptive}}$ | 88.1 | 88.3 |
>   - The main purpose of policy gradient methods such as REINFORCE and PPO is to enable differentiable learning towards potentially non-differentiable metrics. Moreover, the reward is usually a sequence-level delayed one. In contrast, the research objective of EMO focuses on exploring more effective probability divergence measures at the token level through the lens of Earth Mover Distance. In the LLM area, the former usually happens in the alignment stage while EMO is more suitable to be applied to (1) domain-specific fine-tuning; (2) continual pre-training and (3) instruction-tuning. We have included additional results on instruction tuning upon LLaMa2 in the general response. Please refer to it for more details.
> - **Perplexity**: You are right, the evaluation is conducted over the sampled sentences instead of the test set. The $\text{PPL}^{\text{oracle}}$ reported in Table 2 is computed using the Oracle data generator on the model-generated samples. A lower $\text{PPL}^{\text{oracle}}$ indicates higher precision. Here we additionally include the $\text{PPL}^{\text{test}}$ on the test set in the table below:
> |  | $\text{PPL}^{\text{test}}$↓ | $\text{PPL}^{\text{oracle}}$↓ | MAUVE↑ |
> |:---:|:---:|:---:|:---:|
> | MLE | 70.1 | 114.5 | 77.5 |
> | EMO | 74.9 | 55.9 | 83.4 |
> - **Label Smoothing**: The MAUVE scores of label smoothing(smoothing factor set to 0.1) on language modeling tasks are shown below. Label smoothing performs even worse than MLE. The reason is that the number of plausible alternatives to ground-truth tokens is typically much smaller than the number of improbable tokens. Thus,  label smoothing blindly puts more probability mass on low-quality tokens, leading to worse precision.
> |  | WikiText-2 | WritingPrompt | PTB | AG News |
> |---|:---:|:---:|:---:|:---:|
> | MLE | 77.5 | 83.6 | 76.1 | 75.0 |
> | Label Smoothing | 75.8 | 81.6 | 71.9 | 73.8 |

---

> > ### Comment · Reviewer_JZqx · 2023-11-16
> >
> > Thanks for your reply.
> >
> > 1. The ppl test of EMO is much higher than MLE, in my view it seems EMO trades diversity for quality.
> > 2. You said Moreover, the reward is usually a sequence-level delayed one. I think that is not the main difference between PPO/RLHF and EMO. I can also train a word-level one, and obviously the seq-level one is better due to the reward should be signed after the whole sentecne generated instead of internal state.

---

> ### Author Response · Authors · 2023-11-16
>
> Thanks for reading our responses. Regarding your questions, our further responses are as follows:
> - On one hand, the **diversity** of MLE often come at the cost of generting adverse outputs due to its low precision. For language generation systems, high precision is arguably a higher priority since a single bad output can leave a lasting poor impression on the user. On the other hand, EMO's diversity manifests in the range of **good tokens**, which is more **aligned/similar to human text distribution**, as indicated by the Distinct N-gram, MAUVE score, and harmonic mean of forward/reverse cross-entropy.
> - You are right, the reward can also be a word-level one, depending on what the environment can offer. To summarize, the main differences between EMO and RL-based training include: (1) motivation: exploring alternative **probability distance measures** for language modeling v.s. steer language model towards certain aspects (2) formula: weighted combination of gradient of each token in the vocabulary v.s. reward-augmented maximum likelihood of tokens in the output, and (3) application scenario: domain-specific fine-tuning/adaptation, instruction-tuning, and continual pre-training v.s. alignment.

---

> > ### Comment · Reviewer_JZqx · 2023-11-19
> >
> > About the second point, I still can not agree with you. So can you add a section (probably at appendix) to discuss the difference between RLHF and EMO.
> > Please remember I am not mean to criticize EMO, even RLHF is similar to EMO, EMO still has it own advantages (e.g., no need to train an extra rewrad model).

---

> > > ### Author Response · Authors · 2023-11-19
> > >
> > > We are sincerely thankful for your instrumental suggestions. We have included a section to discuss the distinctions between EMO and RLHF in **Appendix A.2**. Specifically, we elucidate the differences from the perspectives of motivation, gradient signal, and application scenarios in great detail.
> > >
> > > Here we would like to briefly explicate the difference in the per time step gradient of EMO and REINFORCE: (1) For EMO, the gradient involves the gradients of **all tokens**' in the vocabulary, weighted by their respective expected transport cost. For REINFORCE, the gradient pertains to the **current token** in the output, weighted by reward computed by a specific reward function(similar spirit shared by policy gradient-based RL); (2) The gradient of EMO is computed directly on $Q_{\theta}$, i.e., $\nabla_{\theta}Q_{\theta}$, while for REINFORCE it is computed based on $\log{Q_{\theta}}$, i.e., $\nabla_{\theta}\log{Q_{\theta}}=\frac{\nabla_{\theta}Q_{\theta}}{Q_{\theta}}$. This "minor" difference actually reflects the essential distinctions between the derivation processes of EMO and policy gradient-based methods.

---

> > > > ### Comment · Reviewer_JZqx · 2023-11-19
> > > >
> > > > I think you are wrong about REINFORCE, because the probability of the token are computed after softmax function, so every token will have gradient.

---

> > > > > ### Author Response · Authors · 2023-11-19
> > > > >
> > > > > Thanks for your reply! We totally agree with you that REINFORCE will penalize the probabilities of tokens other than the target token due to the softmax operation.
> > > > >
> > > > > In the above comment, we use the term "gradient" to refer to the gradient of the explicitly written **loss function**(the one on which you call .backward() in automatic differentiation frameworks like Pytorch) with respect to the model parameters $\theta$. The per time step gradient of the proposed DEMD therefore is $\nabla_{\theta}(\sum_{i=1}^{|V|}Q_{\theta}(v_i)\mathbb{E}_{v_j\sim P}[C(v_i,v_j)])$.
> > > > >
> > > > > For REINFORCE, it is $\nabla_{\theta}(\log{Q_{\theta}(x)}\cdot r)$ for one action(the selection of token $x$) in the sampled trajectory(let us assume an online policy gradient setting), where is $r$ is a scalar that specifies the reward for the current action.
> > > > >
> > > > > Definitely, both algorithms will affect the probabilities of each token in the vocabulary, via different signals.

---

### Official Review · Reviewer_TjsD · 2023-11-01

**Soundness:** 3 good
**Presentation:** 3 good
**Contribution:** 3 good
**Rating:** 6
**Confidence:** 4

**Summary:**

The authors propose to train the language model using an upper bound of Earth Mover Distance. The cost function of EMO is established by the similarity of embeddings from pretrained language model. The authors provide theoretical analysis and argue that EMO is better at handling synonyms compared with MLE. Experiments on various tasks demonstrate EMO's superiority.

**Strengths:**

1. The authors propose to apply the Earth Mover Distance to train the language model and establish an upper bound for practical backward propagation training.
2. The experiments demonstrate promising performance of EMO on various tasks.

**Weaknesses:**

1. The authors argue that MLE exhibits a recall-prioritization behavior, which serves as the primary motivation for introducing their proposed approach, EMO. The claim appears to be confusing, as MLE is equivalent to minimizing the forward KL-divergence, i.e., $KL(p||q_{\theta})$.
If the model $q_{\theta}$ has sufficient capacity, the optimal $q_{\theta}$ converges to $p$. Otherwise, $q_{\theta}$ tends to exhibit a mean-seeking behavior. Therefore I have doubts about whether "recall-prioritization" is proper.

2. As pointed out in Section 3.3, EMO exhibits a property of harmonizing recall and precision. A straightforward inference is that EMO is better at handling synonyms compared to MLE, potentially granting EMO-trained models the capability to generate more diverse texts than models trained with MLE. The authors did not conduct such experiments.

3. I guess the distribution of model trained by EMO is very different from that by MLE. However, there's no experiments and analyses regarding that.

**Questions:**

What are the results of EMO-trained models and MLE-trained models when employing beam search instead of sampling?
I am curious as sampling reflects the entire distribution, whereas beam search captures the distribution's mode.

---

> ### Author Response · Authors · 2023-11-16
>
> Dear reviewer, we genuinely thank you for dedicating your time to reviewing the paper. Here, we strive to effectively respond to the issues you raised.
> - **recall-prioritization**: Conceptually, as described in Section 2.2.1, a high recall means that high likelihood tokens under $P$ shall also have high likelihood under $Q_{\theta}$. In contrast, precision focuses on measuring whether low-quality tokens (unlikely under $P$) have low probabilities under $Q_{\theta}$. Ideally, with unlimited training data and model capacity, as well as a perfect optimizer, MLE-learned distribution will conform to the real data distribution. However, in practice, due to the lack of explicit training signals to penalize low-quality tokens, MLE-learned distribution tends to put non-trivial probability mass over low-quality samples.
>
>
>     The **operationalization of recall and precision** in language modeling has also been investigated by Meister et. al[1]. They quantitiatively measure the recall by forward cross-entropy $\text{CE}^{\text{forward}}=-\sum{P\log{Q_{\theta}}}$, and precision by reverse cross-entropy $\text{CE}^{\text{reverse}}=-\sum{Q_{\theta}\log{P}}$. **The recall-prioritization(low-precision) behavior of MLE can be observed by comparing the original distribution against the distributions modified by various inference-time truncation methods, e.g., top-p, top-k.**
>
>
>     Here we demonstrate this by reporting $\text{CE}^{\text{forward}}$, $\text{CE}^{\text{reverse}}$, and their harmonic mean against a reference distribution $P^{\text{ref}}$ provided by a larger GPT-Neo model. Results on the test set of WikiText-2 and WritingPromptare are shown in the following table. As can be seen, EMO emerges as a training objective that strikes a better balance between recall and precision.
> |  |  | WikiText-2 |  |  | WritingPrompt |  |
> |:---:|:---:|:---:|:---:|:---:|:---:|:---:|
> |  | $\text{CE}^{\text{reverse}}$↓ | $\text{CE}^{\text{forward}}$↓ | Harmonic Mean↓ | $\text{CE}^{\text{reverse}}$↓ | $\text{CE}^{\text{forward}}$↓ | Harmonic Mean↓ |
> | $Q^{\text{one-hot}}$ | 3.229 | 16.83 | 5.154 | 3.316 | 17.742 | 5.326 |
> | $Q_{\theta}^{\text{MLE}}$ | 4.503 | 4.333 | 4.219 | 4.250 | 3.920 | 3.994 |
> | $Q_{\theta}^{\text{EMO}}$ | 3.940 | 4.455 | 3.973 | 3.538 | 4.008 | 3.664 |
> - **diversity**: Though EMO is better at harmonizing recall and precision, it does not directly translate to higher diversity in the model generations. In fact, the MLE-tuned model may generate more diverse outputs because it allocated probability mass more than it should to low-quality outputs. We present the Distinct N-gram($\frac{|\text{unique n-grams}|}{|\text{total n-grams}|}$) of GPT-2-generated continuations on the WikiText-2/WritingPrompt test set in the table below. Compared to MLE, the EMO-tuned model shows n-gram repetitiveness and MAUVE score **closer** to that of human texts by notable margins.
> |  |  |  | WikiText2 |  |  |  |  |  | WritingPrompt |  |  |
> |---|:---:|:---:|:---:|:---:|:---:|---|:---:|:---:|:---:|---|:---:|
> |  |  |  | Dist$_{n}$ |  | MAUVE |  \|  |  |  | Dist$_{n}$ | | MAUVE |
> |  | n=1 | n=2 | n=3 | n=4 | - | \| | n=1 | n=2 | n=3 | n=4 | - |
> | $Q_{\theta}^{\text{MLE}}$ | 0.748 | 0.971 | 0.994 | 0.998 | 77.5 | \| | 0.806 | 0.983 | 0.995 | 0.997 | 83.6 |
> | $Q_{\theta}^{\text{EMO}}$ | 0.698 | 0.955 | 0.990 | 0.997 | 87.5 | \| | 0.762 | 0.972 | 0.993 | 0.996 | 87.4 |
> | Human | 0.687 | 0.958 | 0.988 | 0.996 | 100 | \| | 0.764 | 0.974 | 0.993 | 0.997 | 100 |
> - **Qualitative difference**: In addition to the quantitative difference between MLE and EMO, here we show some predicted next token probabilities from $Q_{\theta}^{\text{MLE}}$ and $Q_{\theta}^{\text{EMO}}$. We show the top-5 predicted tokens along with their probability due to space constraints. EMO tends to a
>     - Prefix: Donald Trump has multiple occupations, such as
>     - $Q_{\theta}^{\text{MLE}}$: the(0.393), a(0.325), construction(0.100), executive(0.093), president(0.088),...
>     - $Q_{\theta}^{\text{EMO}}$: a(0.366), the(0.353), president(0.100), executive(0.091), construction(0.091),...
>     - Prefix: Elephant is the largest living
>     - $Q_{\theta}^{\text{MLE}}$: animal(0.275),creature(0.231),elephant(0.215),mammal(0.197),ant(0.079),...
>     - $Q_{\theta}^{\text{EMO}}$: animal(0.268),creature(0.257),mammal(0.216),elephant(0.196),rept(0.062),...
> - **Beam Search**: We present the MAUVE, BLEU, and ROUGE-1/2/L scores of model outputs generated by beam search(beam width set to 4) on the test set of WikiText-2. With beam search, EMO still displays higher similarity with human texts. For more results, please refer to our general responses where we compare the quality of greedy decoding generations from fine-tuned LLaMa2 models.
> |  | MAUVE | BLEU | ROUGE-1 | ROUGE-2 | ROUGE-L |
> |:---:|:---:|:---:|:---:|:---:|:---:|
> | $Q_{\theta}^{\text{MLE}}$ | 9.93 | 21.31 | 37.0 | 24.7 | 33.6 |
> | $Q_{\theta}^{\text{EMO}}$ | 13.14 | 22.34 | 37.2 | 22.9 | 33.9 |

---

> > ### Comment · Reviewer_TjsD · 2023-11-16
> >
> > Thanks for your reply.
> >
> > I found the analysis of diversity above appears to be contradictory to the motivation of EMO.
> >
> > In the motivation (Sec 2.2.2), the authors contend that **"the ideal training objective should assign high probabilities to those synonyms tokens rather than penalize them as did in MLE"**. To this end, the authors propose EMO, which considers specific tokens as synonyms by assigning them a low transportation cost, leading to a minor penalty for these outputs. Based on its motivation, EMO is anticipated to acquire a more varied distribution (i.e., considering synonyms).
> >
> > However, the authors find EMO leads to a worse diversity and attribute the reason to that **"MLE-tuned model allocated probability mass more than it should to low-quality outputs"**. This is not convincing.

---

> ### Author Response · Authors · 2023-11-16
>
> Thank you for spending time reading our reponse. Regarding your concern, our responses are as follows:
>
> - The number of synonyms/semantically-similar tokens is typically much smaller than that of improbable tokens. Therefore, by explicitly signifying training signal of **low-quality tokens** and the **relative goodness of high-quality tokens**, $Q_{\theta}^{\text{EMO}}$ is less diverse in terms of **full output distribution** because a large portion of probability mass are removed from those low-quality tokens and reassigned to a smaller set of **plausible tokens**. In other words, EMO is more **diverse** in the range of good tokens. Another evidence to show this that, even after applying advanced inference-time sampling strategies(e.g., temperature(T=0.7), top-p(p=0.9), and top-k(k=20)), MLE still underperform EMO because the distribution rescaling behavior of these sampling strategies is token-agnostic. On WikiText-2, the MAUVE score of MLE with top-p and top-k are 81.5, 80.3 respectively.
> - Pursuing diversity in the full-distribution does not entail a human-like distribution. As indicated by the results we reported above, EMO leads to more **human-like** distribution in terms of (1) output N-gram diversity (2) MAUVE score and (3) Harmonic mean of forward/reverse cross-entropy.

---

> > ### Comment · Reviewer_TjsD · 2023-11-19
> >
> > Thanks for your reply.
> >
> > You said **EMO is more diverse in the range of good tokens**. How to prove that? A natural inference is that EMO are better in terms of both quality and diversity. However, it is not the case. Moreover, I don't see the connection between the provided results of MLE and your claim on EMO's diversity.
> >
> > I think this problem is major, as it is directly related to the motivation of this paper.

---

> > > ### Author Response · Authors · 2023-11-20
> > >
> > > Thanks for your reply!
> > >
> > > The main purpose of the results provided above is to demonstrate that the sampled outputs from $Q_{\theta}^{\text{EMO}}$  are more similar to human texts in terms of different metrics, such as Distinct N-gram, MAUVE score, and distribution measures like forward/reverse cross-entropy(and their harmonic mean).
> > >
> > > To illustrate that EMO is more diverse in the range of good tokens, one straightforward way is to calculate the information entropy  $H^{}$ of the normalized top-k probability distribution $Q_{\theta}^{\text{k}}(x_i)=\frac{Q_{\theta}(x_i)}{\sum_{j=1}^{k} Q_{\theta}(x_j)}$, where $x_i$ indicates the $i$-th token in the sorted original distribution $Q_{\theta}$. We calculate the entropy at each token position on the test set of WikiText-2 using GPT-2 and report the results in the table below:
> > >
> > > |  | **k=2** | **k=3** | **k=4** | **k=5** | **k=6** | **k=7** | **k=8** |
> > > |:---:|:---:|:---:|:---:|:---:|:---:|:---:|:---:|
> > > | $Q_{\theta}^{\text{MLE}}$ | 0.561 | 0.874 | 1.090 | 1.254 | 1.382 | 1.488 | 1.747 |
> > > | $Q_{\theta}^{\text{EMO}}$ | 0.571 | 0.885 | 1.110 | 1.258 | 1.373 | 1.476 | 1.737 |
> > >
> > > As we can see, the information entropy of EMO-induced distribution is higher around larger values of k, which means that the EMO-trained model learns to allocate more probability mass to plausible alternatives. As k continues to increase, EMO-induced distribution exhibits a lower entropy because of the lower probability assigned by EMO to the unlikely tokens. We also report the average next-token accuracy on WikiText-2 to show that EMO's diversity in the range of good tokens does not compromise its language modeling accuracy, i.e., the ability to produce an accurate ranking for different tokens.
> > >
> > > |  | **Next Token Accuracy** |
> > > |:---:|:---:|
> > > | $Q_{\theta}^{\text{MLE}}$ | 34.4 |
> > > | $Q_{\theta}^{\text{EMO}}$ | 34.9 |

---

### Official Review · Reviewer_2RWM · 2023-11-08

**Soundness:** 4 excellent
**Presentation:** 4 excellent
**Contribution:** 3 good
**Rating:** 8
**Confidence:** 4

**Summary:**

This paper proposes to train "decoder-only" language model with a different loss function EMO  derived from the EMD (Earth Moving Distance).  The motivation is threefold: Tradeoff between Recall and Precision, Negative Diversity, Train-Test Consistency. These three points are clearly defined and compared for the standard loss (maximum loglikelihood, MLE) and EMO. The authors propose to define the inner cost of this distance with a semantic similarity (cosine between word vectors obtained from a pretrained LM). They also use a more tractable upperbound on the EMD.  The experimental setup proposes different kind of evaluation to assess the impact of this new training strategy.

**Strengths:**

The paper is overall well written and describes an interesting idea. The experimental setup is well described and the results look reproducible.

**Weaknesses:**

A "related work" section is missing, and it could be nice to better discuss the introduction of EMD (and optimal transport in general) in NLP and this kind of task.

The experimental setup focuses on the improvement of MAUVE. This is a quite recent metric that makes a tradeoff between precision and recall. While it is interesting to use that metric, it could be nice also to provide also perplexity. I know MLE optimizes the perplexity so it is not fair for EMO, but it can provides a meaningful comparison point (I mean in table 1).

**Questions:**

The decoding process is unique and since your purpose is to improve diversity, it could be nice to have a discussion on decoding, ie generation.

---

> ### Author Response · Authors · 2023-11-16
>
> Thanks very much for your valuable suggestions and comments. We sincerely appreciate your time in reading the paper, and our point-to-point responses to your comments are given below.
> - **related work section**: Thanks for your advice, we will include a related work section discussing how optimal transport was previously explored and exploited in NLP.
> - **perplexity**: We present additional results of fine-tuned GPT-2's perplexity on the test set of WikiText-2 and WritingPrompts in the following table. In addition, we also report the token-level averaged forward cross-entropy $\text{CE}^{\text{forward}}=-\sum{P\log{Q_{\theta}}}$, reverse cross-entropy $\text{CE}^{\text{reverse}}=-\sum{Q_{\theta}\log{P}}$, and their harmonic mean, computed using the learned model distribution $Q_{\theta}$and the reference distribution $P_{\text{ref}}$ provided by a language model with stronger capacity, i.e., GPT-Neo. These metrics comprehensively delineate the behavior differences between MLE and EMO. Based on the results, $ Q_{\theta}^{\text{EMO}} $ exhibits slightly higher ppl compared to $Q_{\theta}^{\text{MLE}}$. However, EMO excels at striking the balance between recall and precision, resulting in a more human-like distribution.
> |  | Perplexity | MUAVE | $\text{CE}^{\text{forward}}$ | $\text{CE}^{\text{reverse}}$ | Harmonic Mean |
> |---|---|---|---|---|---|
> | $ Q_{\theta}^{\text{MLE}} $ | 29.8 | 77.5 | 4.33 | 4.503 | 4.219 |
> | $ Q_{\theta}^{\text{EMO}} $ | 30.9 | 87.5 | 4.45 | 3.940 | 3.973 |
>
> - **Decoding**: Would you please kindly explain "discussion on decoding, ie generation"? If you are referring to the decoding method, we employ unbiased sampling as our primary decoding method as it allows us to explore the learned distribution in an unbiased way[1,2]. With EMO, we expect the output from the learned model distribution to be more similar to human distribution than MLE. Besides MAUVE, we also include the Distinct N-gram($\frac{|\text{unique n-grams}|}{|\text{total n-grams}|}$) of model-generated continuations on the test set of WikiText-2 and WritingPrompt. The closer to that of human text, the better. As shown in the table, EMO also displays higher similarity in terms of n-gram characteristics.
> |  |  | WikiText-2 |  |   |  | WritingPrompt |  |  |
> |:---:|:---:|:---:|:---:|:---:|:---:|:---:|:---:|:---:|
> |  | n=1 | n=2 | n=3 | n=4 | \| |n=1 | n=2 | n=3 | n=4 |
> | MLE | 0.738 | 0.971 | 0.994 | 0.998 | \| | 0.806 | 0.983 | 0.995 | 0.997 |
> | EMO | 0.698 | 0.955 | 0.990 | 0.997 | \| | 0.762 | 0.972 | 0.993 | 0.996 |
> | Human | 0.687 | 0.948 | 0.988 | 0.996 | \| | 0.764 | 0.974 | 0.993 | 0.997 |
>
> Reference:
>
> [1] Zhang et al.: MIXCE: Training Autoregressive Language Models by Mixing Forward and Reverse Cross-Entropies. ACL 2023
>
> [2] Eikema et al.: Is MAP decoding all you need? the inadequacy of the mode in neural machine translation. ICCL 2020

---

### Author Response · Authors · 2023-11-16
**Additional Results of Instuction Tuning using LLaMa2**

Thanks for the valuable suggestions and comments made by all reviewers. To more comprehensively demonstrate the effect of EMO beyond language modeling and continual pre-training, we also extend our experiments to supervised instruction-tuning upon LLaMa2-7B. We use MLE and EMO to fine-tune LLaMa2-7B on two instruction-tuning datasets, namely [Evol-70K](https://huggingface.co/datasets/WizardLM/WizardLM_evol_instruct_70k) and [OpenPlatypus](https://huggingface.co/datasets/garage-bAInd/Open-Platypus), and utilize [AlpacaEval](about:blank) as the evaluation testbed. We use fine-tuned models to generate responses for each of the 805 test instructions in AlpacaEval using **greedy decoding** and gauge the responses' quality using both close-source API(GPT-4) and open-source judge model([Auto-J](https://github.com/GAIR-NLP/auto-j), fine-tuned from LLaMa2-13B).
The results of the win rate against text-davinci-003 judged by GPT-4 are shown in the table below:

|  | Evol-70K | OpenPlaytpus |
|:---:|:---:|:---:|
| MLE | 76.2 | 58.9 |
| EMO | 78.8 | 63.0 |

Due to the huge cost of GPT-4, we resort to Auto-J, a competitive judge model for testing response quality against a wealth of 7B parameters LLMs of which the responses are publically availabel on AlpacaEval. The results of win rate are shown in the table below:
| Competetor Model | Evol-70K | Evol-70K | OpenPlaytpus |  OpenPlaytpus |
|---|:---:|:---:|:---:|:---:|
|  | MLE | EMO | MLE | EMO |
| Davinci-003 | 91% | 93% | 77% | 78% |
| Baize-v2-7B | 82% | 86% | 57% | 59% |
| LLaMa2-7B-Chat | 63% | 65% | 37% | 39% |
| Vicuna-7B-v1.3 | 76% | 80% | 48% | 52% |
| Zephyr-7B-alpha | 70% | 72% | 40% | 44% |

The above results demonstrate the EMO\-tuned models are able to generate responses deemed more helpful by both powerful GPT-4 and open-source judge model like Auto\-J, manifesting its theoretic advantages(harmonizing recall and precision, negaitve diversity awareness, and better train-test consistency) over MLE.

---

### Author Response · Authors · 2023-11-18
**Updated manuscript**

Dear reviewers, we have updated our manuscript. Thanks for your constructive comments and valuable suggestions. This new version includes updated appendix with additional experiments of instruction-tuning upon LLaMa series and necessary language modeling results as suggested by the reviewers. If you have any further concerns, please let us know and we are more than willing to address them.

---

### Author Response · Authors · 2023-11-22
**Additional Results of Instruction-tuned LLaMa2-7B on Common Benchmarks**

In addition to the subjective evaluation conducted using AlpacaEval, we also evaluate using conventional objective benchmarks, including MMLU, Big Bench Hard, and GSM. Specifically, we fine-tune the pre-trained LLaMa2-7B-base model with MLE/EMO on the mixture of the [chain-of-thought submix of Flan-v2](https://beaker.org/api/v3/datasets/01GXZ52K2Q932H6KZY499A7FE8/files/cot_zsopt.jsonl) and [unnatural instruction](https://github.com/orhonovich/unnatural-instructions/raw/main/data/core_data.zip), keeping all training hyperparameters(max length, learning rate, epochs, etc.) the same. The results are presented in the table below. Combined together, we can see that the benefit of a more accurate distribution induced by EMO leads to quality improvement in both open-ended generation and better language understanding capability.

|  | **MMLU** | **BBH** | **GSM** |
|:---:|:---:|:---:|:---:|
| MLE | 46.3 | 36.7 | 31.0 |
| EMO | 47.7 | 37.3 | 36.0 |

---

### Meta-Review · Area_Chair_jaib · 2023-12-06

**Metareview:**

The paper proposes training language models using an upper bound of Earth Mover Distance, which is based on embedding similarity. The paper claims that this can address some of the practical limitations of LMs trained with MLE for text generation, in particular "recall-prioritization" due to assigning too much weight to the ground truth next token, and without explicitly recognizing that other words in the vocabulary are not equally incorrect. Results on Llama models with continual fine-tuning on WikiText-103 shows performance improvements across various NLU tasks, while fine-tuning GPT-2 and OPT-125M leads to improved performance on open-ended text generation evaluation metrics such as Mauve scores (with pure sampling).

The main strengths of the paper are the novelty of the method and strong experimental results. The main weakness is that it is not clear enough that the paper's hypotheses about the weaknesses of MLE (prioritizing recall and ignoring negative diversity) compared to the proposed EMO approach (learning a better distribution over plausible tokens) are supported by the theoretical properties of the objectives or by the experimental results. However, given that the question about what properties a good objective function for training models for text generation has not been settled yet, the paper's contribution in terms of proposing a new method and hypotheses may still be a valuable addition to the literature on this question. Therefore while the authors should update the paper to address the questions raised by the reviewers and be more careful about the claims that are made, on balance the paper should still be accepted.

**Justification For Why Not Higher Score:**

The method makes a clear contribution, but there are some unresolved questions regarding the framing of the paper.

**Justification For Why Not Lower Score:**

While some of the reviewers were not convinced about the motivation of the proposed method and whether the paper's hypotheses are sufficiently supported by the experimental results, there is still a clear contribution to the debate about training objectives for language models and sufficient positive results to justify acceptance.

---

### Decision · Program_Chairs · 2024-01-16

Accept (poster)